# CRISPR/Cas with ribonucleoprotein complexes and transiently selected telomere vectors allows highly efficient marker-free and multiple genome editing in *Botrytis cinerea*

Thomas Leisen[1], Fabian Bietz[1], Janina Werner[1], Alex Wegner[2], Ulrich Schaffrath[2], David Scheuring[1], Felix Willmund[1], Andreas Mosbach[3], Gabriel Scalliet[3], Matthias Hahn[1]*

1 University of Kaiserslautern, Department of Biology, Kaiserslautern, Germany, 2 RWTH Aachen University, Department of Plant Physiology, Aachen, Germany, 3 Syngenta Crop Protection AG, Stein, Switzerland

* hahn@biologie.uni-kl.de

**Data Availability Statement:** All relevant data are within the manuscript and its Supporting Information files.

## Abstract

CRISPR/Cas has become the state-of-the-art technology for genetic manipulation in diverse organisms, enabling targeted genetic changes to be performed with unprecedented efficiency. Here we report on the first establishment of robust CRISPR/Cas editing in the important necrotrophic plant pathogen *Botrytis cinerea* based on the introduction of optimized Cas9-sgRNA ribonucleoprotein complexes (RNPs) into protoplasts. Editing yields were further improved by development of a novel strategy that combines RNP delivery with cotransformation of transiently stable vectors containing telomeres, which allowed temporary selection and convenient screening for marker-free editing events. We demonstrate that this approach provides superior editing rates compared to existing CRISPR/Cas-based methods in filamentous fungi, including the model plant pathogen *Magnaporthe oryzae*. Genome sequencing of edited strains revealed very few additional mutations and no evidence for RNP-mediated off-targeting. The high performance of telomere vector-mediated editing was demonstrated by random mutagenesis of codon 272 of the *sdhB* gene, a major determinant of resistance to succinate dehydrogenase inhibitor (SDHI) fungicides by in bulk replacement of the codon 272 with codons encoding all 20 amino acids. All exchanges were found at similar frequencies in the absence of selection but SDHI selection allowed the identification of novel amino acid substitutions which conferred differential resistance levels towards different SDHI fungicides. The increased efficiency and easy handling of RNP-based cotransformation is expected to accelerate molecular research in *B. cinerea* and other fungi.

## Author summary

In this study, we describe the establishment of the CRISPR/Cas technology for genome editing in the gray mold fungus *Botrytis cinerea*, one of the economically most important plant pathogens worldwide. We report the development of a strategy which combines the

**Funding:** This work was supported by Syngenta and the BioComp Research Initiative of Rhineland-Palatinate, Germany. Alex Wegner was supported by a PhD grant of RWTH Aachen University. The funders had no role in study design, data collection and analysis, decision to publish, or preparation of the manuscript.

**Competing interests:** The authors have declared that no competing interests exist.

introduction of an optimized nuclear-targeted Cas9-single guide RNA ribonucleoprotein complex (RNP) and a repair template together with unstable telomere vectors for transient selection into fungal protoplasts. A high proportion of the transformants contains the desired genetic changes, and the telomere vector is lost subsequently when selection is stopped. This system allowed introduction of changes into the genome without the requirement of selection markers. It shows superior editing efficiencies compared to existing CRISPR/Cas protocols for filamentous fungi, and leads to a very low number of additional off-target mutations. To demonstrate the performance of our protocol, we conducted for the first time a site-directed, random mutagenesis in a gene encoding an important fungicide target. This approach allows new applications such as *in vivo* structure-function analysis of proteins and rational fungicide resistance studies. As demonstrated with the rice blast pathogen *Magnaporthe oryzae*, the RNP-based CRISPR/Cas toolset with telomere vectors can be transferred to other fungi and is expected to boost their genetic manipulation.

## Introduction

*Botrytis cinerea* is a plant pathogenic ascomycete which infects more than thousand species, causing gray mold disease which is responsible for over a billion dollars of losses in fruits, vegetables and flowers every year [1]. Due to its worldwide occurrence, great economic importance and non-specific necrotrophic lifestyle, it has been ranked as the second most important plant pathogenic fungus [2]. Control of gray mold often requires repeated treatments with fungicides, in particular under high humidity conditions, but rapid adaption and resistance development has dramatically reduced their efficiency worldwide in many field and greenhouse crops, for example in strawberries [3]. Since the succinate dehydrogenase inhibitor fungicides (SDHI), no new major antifungal modes of action have been released for the control of *Botrytis* in the last years [4]. Within this fungicidal class, new molecules are developed which display higher intrinsic activities, better physicochemical properties and reduced risk compared to existing solutions. After germination of *B. cinerea* conidia on the plant surface, the fungus penetrates and invade the host tissue, rapidly killing plant cells by releasing a complex mixture of cell wall degrading enzymes, phytotoxic metabolites and proteins, and by host tissue acidification [5, 6]. How host cell death is induced is not fully understood, but several lines of evidence indicate that the invading hyphae trigger the hypersensitive response, a plant-specific type of programmed cell death linked to strong defense reactions [7, 8]. Furthermore, *B. cinerea* releases small RNAs (sRNAs) that can suppress the expression of defense-related genes in its host plants [9]. Reciprocally, plants also release sRNAs aimed to suppress fungal virulence [10]. To facilitate access to genes or non-coding RNA loci that are important for pathogenesis, a gapless genome sequence of *B. cinerea* has been published [11]. Considerable efforts have been made to generate improved tools for the genetic manipulation of *B. cinerea*. *Agrobacterium*-mediated and protoplast-based transformation have been developed [12–14], and several vectors are available which facilitate the generation of mutants and strains expressing fluorescently tagged proteins for cytological studies [15]. Nevertheless, generation of mutants remains time-consuming, partly because of the multinuclear nature of *B. cinerea*, which demands several rounds of sub-cultivation on selective media to achieve homokaryosis. Furthermore, the generation of multiple knock-out mutants was hampered until now by the lack of marker recycling systems for serial gene replacements, as described in some filamentous fungi [16].

The application of the clustered regularly interspaced short palindromic repeats (CRISPR)-associated RNA-guided Cas9 endonuclease activity has revolutionized genome editing and greatly facilitated the genetic manipulation in a wide range of species [17]. CRISPR/Cas is based on the introduction of double stranded breaks by the Cas9 endonuclease in the genome of an organism. Cas9 targeting occurs by complementary sequences of a single guide RNA (sgRNA), which directs the endonuclease to a genomic target sequence via a 20 bp homology region [18–20]. The additional sequence requirement, for Cas9 from *Streptococcus pyogenes*, is the presence of the so-called protospacer adjacent motif (PAM), a triplet NGG located immediately 3′ of the target [21]. The breaks are then repaired by non-homologous DNA end joining (NHEJ) which can introduce small insertions or deletions. When a repair template (RT) homologous to sequences flanking the break is provided, repair can also occur by homologous recombination (HR), which allows the generation of specific edits in the genome.

CRISPR/Cas has been successfully applied in various fungal species using different strategies to deliver Cas9 and the sgRNA [22]. In most cases, codon optimized versions of Cas9 encoding genes were introduced by stable chromosomal integration or transiently via plasmids. To achieve robust expression and efficient nuclear targeting of Cas9, strong fungal promoters, codon-optimized genes and suitable nuclear localization signals fused to the protein are beneficial. Delivery of sgRNAs can be achieved either via plasmids or by *in vitro* synthesized sgRNA. More recently, transformation with Cas9-sgRNA ribonucleoprotein (RNP) complexes has been established in several fungi [23–25]. In *M. oryzae*, a novel selection strategy was developed based on switching between two antagonistic states of resistance (either to the fungicides benomyl or diethofencarb), depending on the sequence of codon 198 of the *TUB2* gene, to allow the construction of marker-less mutations [24].

In this study, we show that CRISPR/Cas-based genome editing is highly efficient in *B. cinerea* with Cas9-sgRNA RNPs introduced into protoplasts. By using *Bos1* as a selectable marker for gene inactivation, high frequencies of edits via NHEJ and HR were observed. With RT containing only 60 bp homology flanks, >90% targeting efficiency was achieved. Taking advantage of a transiently selectable telomere vector and high cotransformation rates of RNP constructs, a marker-free editing strategy was developed, yielding up to thousands of edited transformants per transformation. The power of this approach, which worked also well for *Magnaporthe oryzae*, was demonstrated by the one-step random mutagenesis of a resistance-associated codon in a fungicide target gene followed by characterization of mutant collections *in vivo*, an application which has not been possible before in filamentous fungi.

## Results

### Establishment and characterization of CRISPR/Cas editing in *B. cinerea*

To achieve strong expression and robust nuclear localization of Cas9, we tested Cas9 constructs with different nuclear localization signals (NLS) using *B. cinerea* transformants expressing a GFP-tagged synthetic Cas9 gene adapted to the low GC content of *B. cinerea* [26]. In several fungi, Cas9 versions containing a single SV40 T antigen NLS have been found to be active [22]. However, *B. cinerea* transformants expressing Cas9-GFP with a single C-terminal SV40 T antigen NLS, or with two N- and C-terminal SV40 NLS, only resulted in fluorescence distributed between cytoplasm and nuclei (Fig 1A and 1B). Multiple tandem NLS copies have been described to increase nuclear targeting efficiency [27]. We therefore tested the performance of four tandem copies of SV40 NLS (SV40$^{x4}$) and a duplicated NLS of the nuclear *B. cinerea* StuA protein (Stu$^{x2}$), and found that both NLS versions effectively directed Cas9 into nuclei of *B. cinerea* transformants (Fig 1C and 1D).

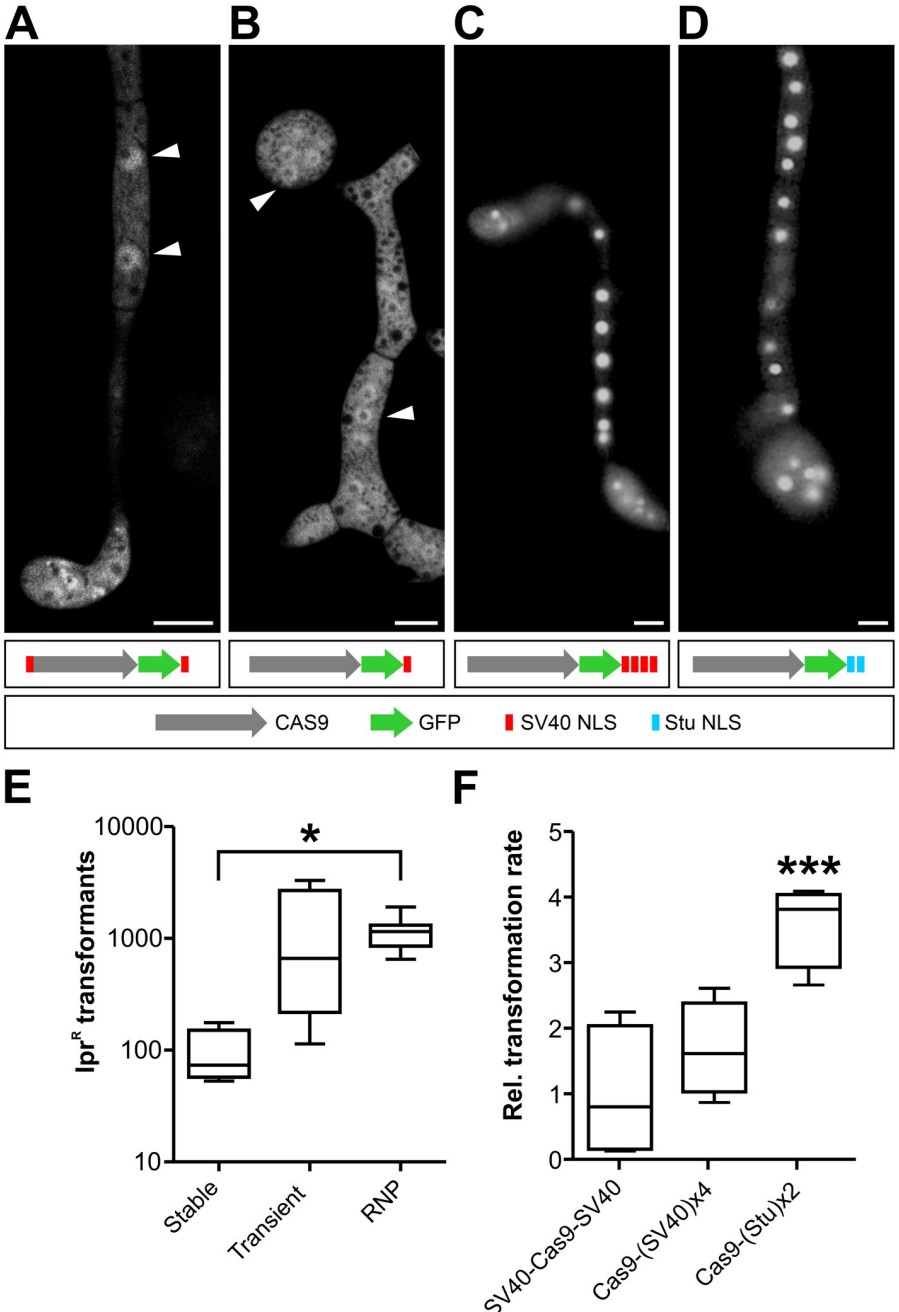

**Fig 1. Optimization of Cas9 nuclear targeting and delivery into *B. cinerea* protoplasts.** (A-D) Subcellular localization of genetically delivered Cas9-GFP constructs fused to different NLS. Fluorescence microscopy images of 18 h old germlings on glass slides. Arrowheads depict nuclei. Only in (C) and (D), fluorescence is concentrated in the nuclei. Scale bars: 5 μm. (E) Transformation rates (NHEJ-mediated, Ipr$^R$ *Bos1* k.o. transformants) obtained in *B. cinerea* with different Cas9 delivery strategies. Cas9 was expressed from a chromosomally integrated gene (Cas9-SV40$^{x4}$-NLS; stable), transiently from a gene on a telomere vector (Cas9-GFP-SV40$^{x4}$-NLS; transient) or added as a protein (Cas9-Stu$^{x2}$-NLS; RNP) together with Bos1-T2 sgRNA to *B. cinerea* protoplasts. The *p* values by one-way ANOVA followed by Tukey's multiple comparisons post hoc test, relative to stable Cas9 expression, are indicated. $^*p \leq 0.05$; stable (n = 4), transient (n = 4), RNP (n = 11). (F) Comparison of different NLS arrangements on genome editing efficiency of Cas9-sgRNA RNPs targeting *Bos1*. Values are relative to transformation rate with SV40-Cas9-SV40. In (E) and (F), no Ipr$^R$ colonies were obtained without Cas9. The *p* values by one-way ANOVA followed by Dunnett's multiple comparisons post hoc test are indicated. $^{***}p$ value $\leq 0.001$; n = 4.

We next tested which strategy was best suited for Cas9 delivery into *B. cinerea* protoplasts. For stable expression, a construct constitutively expressing Cas9-SV40$^{x4}$ was integrated into the *niaD* region of the genome. For transient expression, Cas9-GFP-Stu$^{x2}$ cloned into an autonomously replicating telomere vector (see below) was transformed together with a sgRNA into wild type (WT) *B. cinerea* [28]. Stable and transient expression of Cas9 was confirmed by immunoblot analysis (S1 Fig). Alternatively, purified Cas9-Stu$^{x2}$ protein assembled with a sgRNA to a ribonucleoprotein complex (RNP) were used for transformation of *B. cinerea* WT. CRISPR/Cas activity was evaluated by quantification of error-prone repair via NHEJ, using the *Bos1* gene as a target. *Bos1* encodes a histidine kinase that regulates high osmolarity adaptation via the mitogen activated protein kinase Sak1 [29]. *Bos1* loss-of-function mutants have been shown to be resistant against the fungicides iprodione (Ipr) and fludioxonil (Fld) [30, 31]. This phenotype allows for robust positive selection of *Bos1* mutants generated by NHEJ. With transiently expressed Cas9-GFP-Stu$^{x2}$ and with Cas9-Stu$^{x2}$ RNPs, high numbers of transformants were obtained, whereas stably expressed Cas9-SV40$^{x4}$ yielded significantly fewer colonies (Fig 1E). All transformants tested were both Ipr$^{R}$ and Fld$^{R}$, and failed to produce sporulating aerial hyphae. Compared to the WT, growth of the transformants was more strongly inhibited on media with high osmolarity, and their virulence was strongly reduced when inoculated on tomato leaves (S2 Fig). These phenotypes are consistent with those reported for *Bos1* null mutants [30], and they confirmed that *Bos1* was inactivated in the transformants. Because of the high, reproducible transformation rates obtained, RNP-mediated transformation was used in all subsequent experiments. We then tested recombinant Cas9 protein variants carrying different NLS for their efficiency in RNP-mediated transformation. Compared to a commercially available Cas9 containing SV40-NLS on both termini, *in vivo* editing activities of Cas9 with four C-terminal tandem copies of SV40-NLS were similar, and of Cas9 with two C-terminal Stu-NLS four-fold higher (Fig 1F). Cas9-Stu$^{x2}$-NLS was therefore used for RNP formation in the following experiments with *B. cinerea*.

To further characterize CRISPR/Cas editing via NHEJ, RNPs complexed with different sgRNAs targeting *Bos1* between codons 344 and 372 were introduced into *B. cinerea* protoplasts (Fig 2 and S3 Fig). Variable editing frequencies were obtained, which correlated only weakly with *in silico* predictions and *in vitro* cleavage assays (S3 Fig). RNP-induced *Bos1* mutations in Ipr$^{R}$ transformants were characterized by sequencing. Most of the Ipr$^{R}$ transformants had insertions or deletions of only one or a few nucleotides at the cleavage site, which is typical for error-prone NHEJ repair of CRISPR/Cas-induced DNA breaks [32]. With three sgRNAs, a '+T' insertion was the predominant mutation, while another sgRNA (Bos1-T7) yielded mostly three types of 9 bp deletions (Fig 2). Insertions of >1 bp where found in only 19% of the transformants analyzed. Among 14 insertions of 15–164 bp, three contained *Bos1* DNA derived from sequences close to the sgRNA target sites, and one contained mitochondrial DNA (S4 Fig). The majority of insertions were derived from the scaffold used for sgRNA synthesis, which had apparently resisted the DNase treatment after synthesis and was cotransformed with the RNP. Several more complex insertions were observed that involved amplification of neighboring *Bos1* sequences (S4 Fig). Taken together, our results show diverse NHEJ error-prone repair events as expected, but RNPs with three sgRNAs containing 'T' at position 4 counted from the 3'-terminus, resulted in the same '+T' mutation in 68.3 ± 6.9% of the tested transformants.

## Targeted Cas9-RNP-mediated editing

To generate targeted *B. cinerea* insertion mutants, and to compare NHEJ- and HR-editing frequencies, a fenhexamid resistance cassette (Fen$^{R}$) [33] flanked by 1 kb *Bos1* sequences was

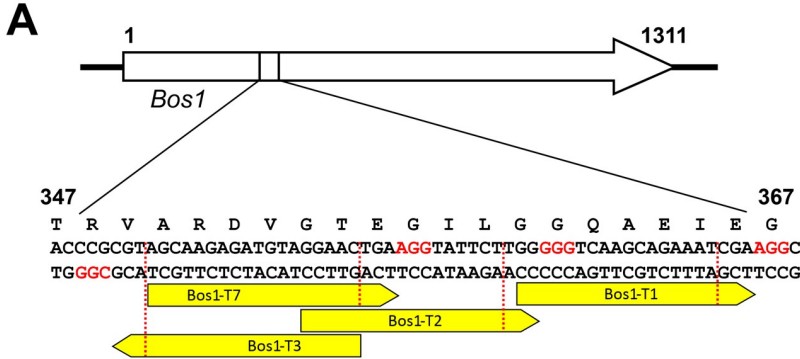

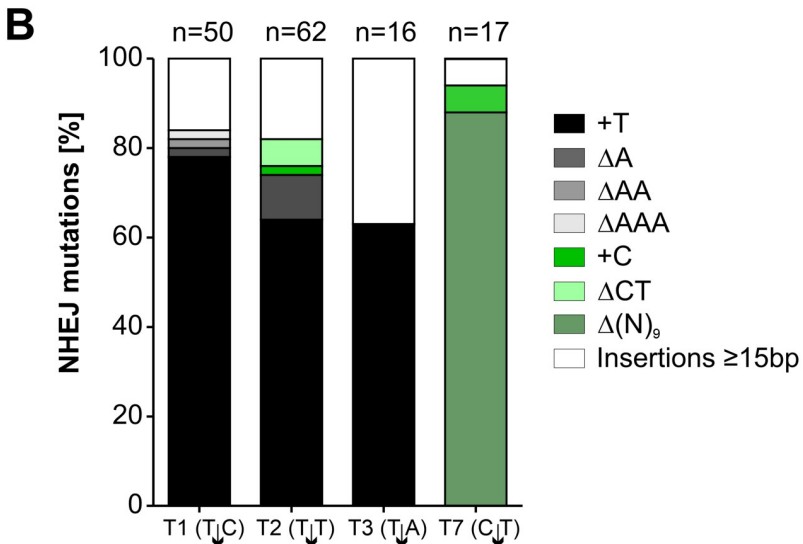

**Fig 2. NHEJ-mediated mutations induced in *Bos1* by RNP mediated genome editing.** (A) Positions of the sgRNAs targeting *Bos1*. (B) Distribution of mutations detected in iprodione resistant transformants. Note that sgRNAs introducing T↓N cleavage sites resulted in a majority of '+T' insertions.

delivered as a repair template (RT) in addition to the Cas9-RNP targeting *Bos1* into protoplasts. Transformants were selected for Fen[R] or Ipr[R]. Regardless whether the RT was provided as circular plasmid or as PCR product, transformation rates appeared to be lower with selection for Fen[R] than with selection for Ipr[R] (S5 Fig). This result can be explained if most of the Fen[R] transformants were the result of HR-mediated integration of the RT into *Bos1*, and the Ipr[R] transformants the result of both error-prone NHEJ repair and HR-mediated RT integration. In agreement with this interpretation, almost all Fen[R] transformants were also Ipr[R], indicating highly efficient HR-mediated integration of the Fen[R] cassette into *Bos1*. In contrast, only 22–39% of Ipr[R] transformants were Fen[R], indicating a 2.5 to 5-fold higher frequency of NHEJ compared to HR in this experiment (S5 Fig).

Conventional gene targeting in filamentous fungi requires resistance cassettes with ≥0.5–1 kb flanking homology regions. A major advantage of CRISPR/Cas is that dsDNA repair via HR can also be achieved using short RT homology flanks [24, 34–36]. To test this for *B. cinerea*, Fen[R] cassettes with *Bos1* homology flanks adjacent to the PAM sequence, ranging from 0 to 60 bp, were generated as RT. When delivered as Cas9-RNPs, the numbers of Fen[R]

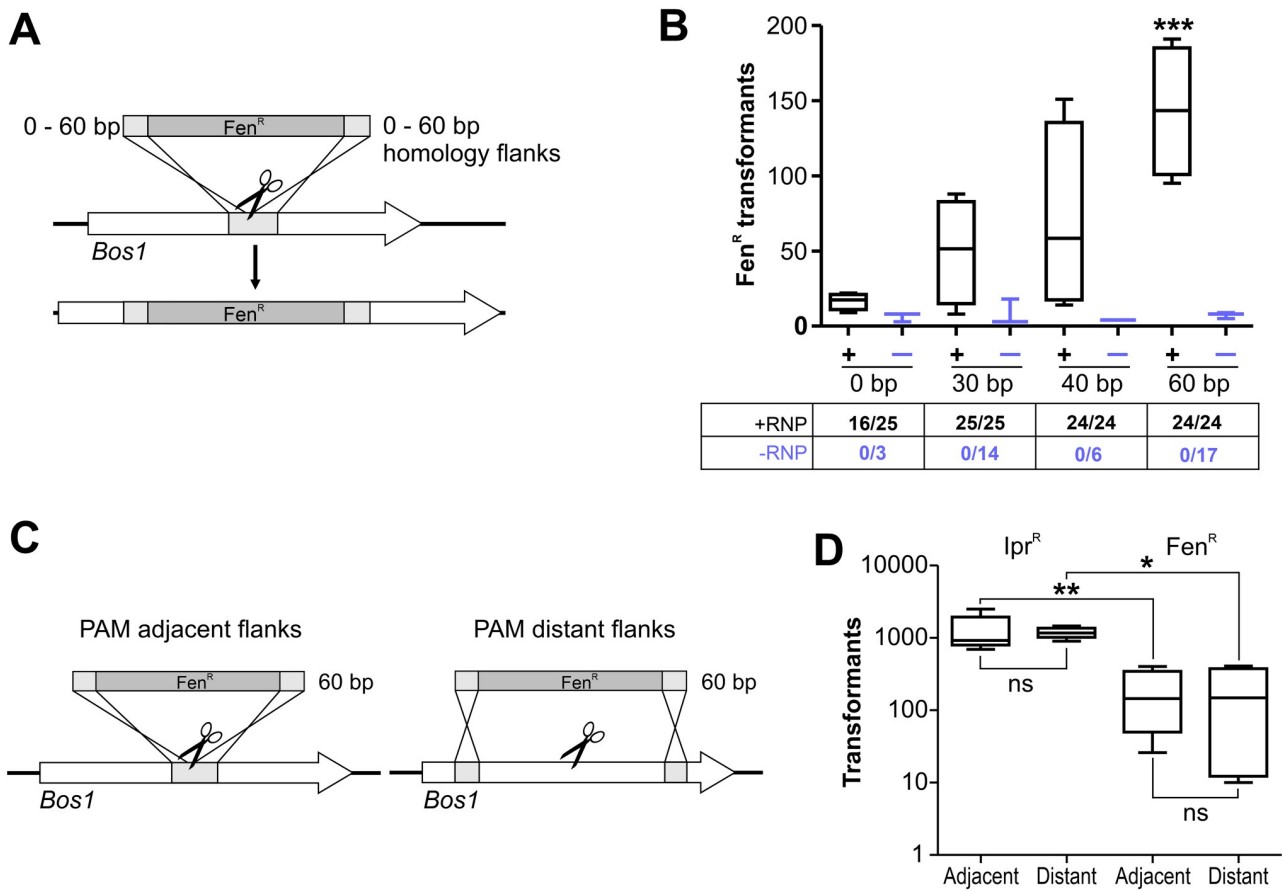

**Fig 3. Efficiency of CRISPR/Cas editing of *Bos1* using repair templates with short homology flanks.** (A) Experimental scheme. (B) Results of transformations with RNP (black lines) and without RNP (blue lines) for RT with flank sizes of 0 to 60 bp. (+RNP: n = 4; -RNP: n = 3). The numbers below show fractions of Fen^R transformants being Ipr^R, indicating targeting efficiencies. The p values by one-way ANOVA followed by Dunnett's multiple comparisons (control: 0 bp) post hoc test are indicated. ***p ≤ 0.001. (C) Scheme of *Bos1* targeting with different placement of 60 bp homology flanks of RT, resulting in insertion (left) or 2 kb deletion (right). (D) Results of transformations with RNP and two types of RT as shown in (C). n = 5 (Ipr^R); n = 4 (Fen^R). The *p* values by one-way ANOVA followed by Tukey's multiple comparisons post hoc test are indicated. *p ≤ 0.05; **
p ≤ 0,01.

transformants increased with increasing flank sizes, reaching highest values with 60 bp flanks (Fig 3A and 3B). All Fen^R transformants tested were Ipr^R, indicating correct targeting of *Bos1*. To verify precise insertion of the Fen^R cassette into *Bos1* by HR, the Fen^R cassette was amplified from several transformants with flanking *Bos1* primers, and the insertion borders were sequenced. Correct insertions were confirmed for two out of three insertions with 30 bp borders, four out of four with 40 bp borders, and two out of three insertions with 60 bp borders. Remarkably, even 64% of the transformants obtained with a Fen^R cassette lacking homology flanks were also Ipr^R. Sequencing revealed in two cases perfect insertions of the Fen^R cassette, in the same orientation as with the *Bos1* borders, and in one case an insertion accompanied by deletion of *Bos1* DNA, which was not analyzed further. When the Fen^R cassettes were delivered without RNP, only few Fen^R transformants were obtained, and none of them were Ipr^R, indicating integrations outside *Bos1* (Fig 3B). When the 60 bp flanks of the RT were separated by 1 kb each from the PAM site to generate a 2 kb *Bos1* deletion instead of an insertion, similar transformation efficiencies were obtained as with adjacent 60 bp flanks (Fig 3C and 3D). Thus,

CRISPR/Cas allows the use of short homology flanks in a flexible way for highly efficient gene targeting.

To exploit the efficiency of CRISPR/Cas, co-targeting of two genes encoding key enzymes for biosynthesis of the phytotoxins botrydial (*bot2*) and botcinic acid (*boa6*) was tested. The role of these toxins for *B. cinerea* is not yet completely clear. Whereas single *bot2* and *boa6* knockout mutants did not reveal a decreased pathogenicity, double mutants were found to be impaired in virulence on bean leaves [37]. Cas9-RNPs and RTs with 60 bp flanks targeting *bot2* (using a Fen$^R$ cassette) and *boa6* (using a cyprodinil (Cyp$^R$) cassette) were generated. In two transformations, 39 and 47 Fen$^R$ colonies, and 16 and 14 Cyp$^R$ colonies, respectively, were obtained (S6A Fig). Of 70 Fen$^R$ transformants tested, 49 were Cyp$^R$, indicating coediting of *bot2* and *boa6*. PCR-based DNA analysis of 20 Fen$^R$ Cyp$^R$ transformants revealed 15 transformants as *boa6* k.o., four as *bot2* k.o., and three as *boa6bot2* double k.o., two of which could be purified to homokaryosis (S6B Fig). Thus, double knock-outs can be obtained with Cas9-RNPs with high frequency, but coediting resulted in increased rates of ectopic integration of the RT. Genome sequencing of a *boa6bot2* transformant (ΔΔ6) confirmed correct homologous insertion of the Cyp$^R$ cassette into *boa6*, and of the Fen$^R$ cassette into *bot2*. Phenotypical characterization of the double mutants revealed no significant differences to the WT in their vegetative growth and infection (S6C and S6D Fig). This result indicated that the phytotoxins botrydial and botcinic acid are not important for *B. cinerea* to infect tomato leaves.

## Resistance marker shuttling, a simple strategy for marker-free editing

To generate precise and multiple changes in the genome, marker-free editing is required. Two marker-free mutagenesis strategies were developed, both exploiting the high efficiency of cotransformation, namely that two or more DNA constructs are taken up by fungal cells with much higher frequencies than expected from single transformation rates. The first strategy, called resistance marker shuttling, is based on the integration of an RT into a non-essential genomic locus in exchange for an existing resistance cassette with identical promoter and terminator sequences which serve as homology flanks. To test for marker exchange, a *B. cinerea* strain carrying a nourseothricin (Nat$^R$) cassette in the *xyn11A* locus [38] was transformed with Cas9-RNP targeting the Nat gene and a Fen$^R$ RT which shared the promoter (PtrpC) and the terminator (TniaD) sequences with the targeted *nat1* gene as homology flanks (Fig 4A). Transformations resulted in several hundred Fen$^R$ colonies, and the majority of them had lost Nat$^R$ as expected for a marker exchange. When to the transformation assay containing Cas9- RNP targeting Nat$^R$ and a Fen$^R$ RT a Cas9- RNP targeting *Bos1* was added, similar numbers of Fen$^R$ transformants were obtained, and 56–74% of them were also Ipr$^R$, demonstrating a high rate of NHEJ coediting. No marker exchange was observed when the Fen$^R$ RT was transformed without Cas9-RNP as negative control (Fig 4B). To test the stability of both resistance markers in the Fen$^R$ Ipr$^R$ double transformants, each ten of them were transferred three times to ME agar plates containing only Fen or Ipr. All transformants treated this way retained the non-selected resistance, indicating that coediting had occurred in the same nuclei of the transformed protoplasts. The resulting transformants could be used for another round of marker shuttling, now targeting the Fen$^R$ resistance cassette.

## Use of transiently selected telomere vectors for completely marker-free editing

Previous studies have shown that plasmids containing a pair of telomeres (pTEL) can be efficiently transformed into filamentous fungi and replicate there autonomously as centromere-free minichromosomes, but are rapidly lost in the absence of selection pressure [28]. To

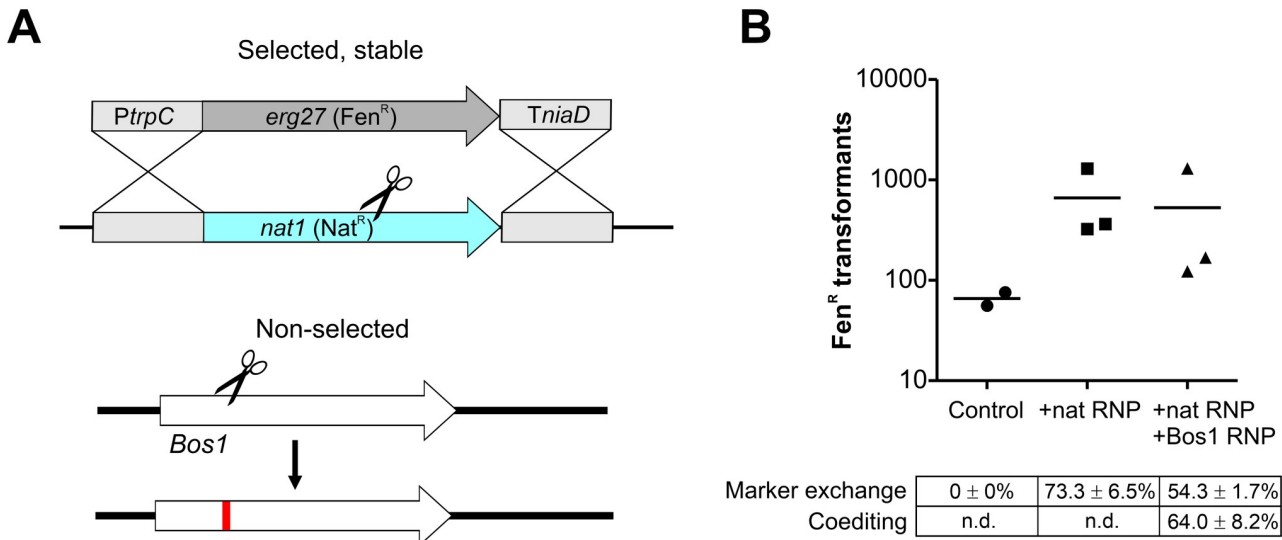

**Fig 4. Application of marker exchange for non-selected CRISPR/Cas mutagenesis of *Bos1* via NHEJ.** (A) Experimental scheme. (B) Results of transformation of *B. cinerea xyn11A*-Nat[R] with Cas9-RNP targeting *nat* using Fen[R] repair template, with or without Cas9-RNP targeting *Bos1*, as shown in (B). Control: Fen[R] repair template added alone.

confirm that a circular pTEL plasmid is converted into a linear chromosome with telomere ends after transformation into *B. cinerea*, and that it is completely lost in edited strains that were constructed via pTEL-mediated editing, genomic DNA of several *B. cinerea* strains was inspected for the presence or absence of pTEL-Fen sequences. A region covering part of the *Aspergillus nidulans trpC* promoter (PtrpC) was detected by PCR in three strains transformed with pTEL-Fen and in the *bot2boa6-6* mutant which contains PtrpC sequences independent of pTEL-Fen integrated in the genome, but not in the WT and several other edited strains. In contrast, the kanamycin resistance gene sequence, which was expected to be lost after pTEL linearization in *B. cinerea*, did not amplify by PCR in any of these strains (S7A and S7B Fig). Full genome sequencing reads from pTEL-Fen-8 transformants maintained onto selection media showed low coverage of sequence reads aligning to the linearized plasmid, indicating a lower than chromosomal copy number of this autonomous genetic element. Furthermore, very few reads were detected that mapped to the telomeric ends, suggesting that shortening of the telomeres might occur *in vivo* (S7C Fig). Taken together, these results confirm the linearization of pTEL-Fen in *B. cinerea*, and its complete loss after stop of selection.

To further test the stability of pTEL plasmids in *B. cinerea* and to observe the speed of their loss in the absence of selection, transformants containing pTEL-Fen, pFB2N, and pTEL-Bc-Cas9GFP-NLS-Stux2 were cultivated on ME+Fen or ME+Hyg plates for selection, and then transferred to a non-selective ME plate. After one week of incubation, conidia that had developed above the inoculation site, and at a distance of 4 cm, were harvested and tested for growth on selective media. For all strains carrying pTEL vectors, rapid loss of resistance was observed during growth on non-selective medium, in contrast to strains with chromosomally integrated resistance markers which showed stable resistance (S1 Table). Based on these properties of telomere vectors, a strategy for cotransformation of a pTEL vector and Cas9-RNP with or without RT was developed, to achieve marker-free CRISPR/Cas editing. This strategy involves the following steps (Fig 5A and 5B): i) cotransformation of pTEL and Cas9-RNP (with or without RT) into *B. cinerea*; ii) selection for pTEL-encoded resistance; iii) identification of transformants with desired editing events; iv) purification of the transformants by transfers on selective media

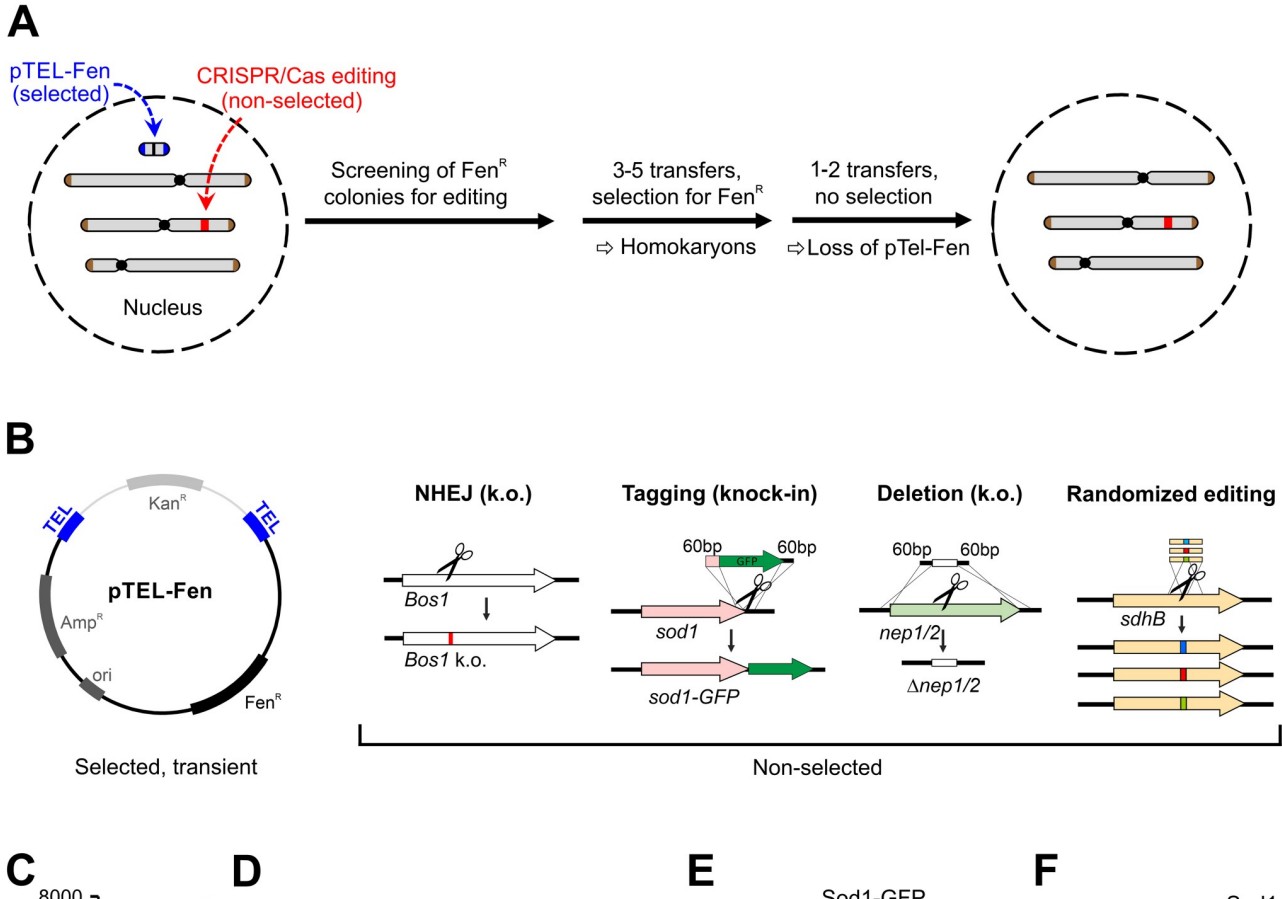

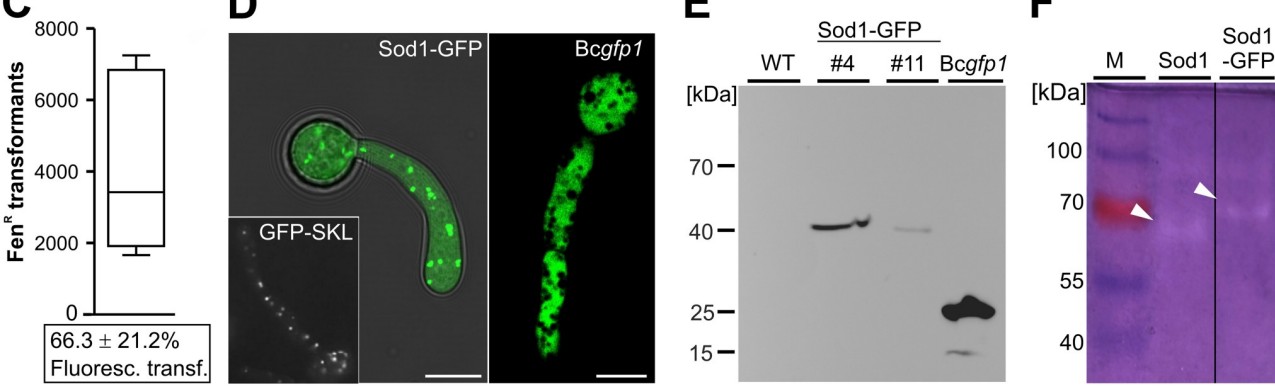

**Fig 5. Telomere vector-mediated editing for introduction of marker-free CRISPR/Cas edits into *B. cinerea*.** (A) Experimental setup. (B) Applications of non-selected editing performed in this study. pTEL-Fen can be propagated in *E. coli* with selection for ampicillin (Amp$^R$) and kanamycin (Kan$^R$). After transformation into *B. cinerea* it is linearized to a minichromosome with telomere ends. (C-F) Generation and characterization of a Sod1-GFP knock-in mutant. (C) Transformation efficiency and percentage of fluorescent transformants (below; n = 3). (D) Cytoplasmic and putative peroxisomal localization of Sod1-GFP fluorescence, as indicated by similarity to the fluorescence pattern of a mutant expressing GFP fused to a SKL peroxisomal targeting motif, and strain Bc*gfp1* expressing GFP only [15]. (E) Immunoblot detection of Sod1-GFP with GFP antibodies. Loaded are *B. cinerea* WT, two transformants of Sod1-GFP mutant, and a strain (Bc*gfp1*) expressing cytoplasmic GFP only. Transformant Sod1-GFP #4 was used also in 5D and 5F. (F) Native gel electrophoresis of *B. cinerea* protein extracts stained for superoxide dismutase activity (inhibition of superoxide-mediated reduction of nitroblue tetrazolium). Lanes showing WT (expressing Sod1) and mutant (expressing Sod1-GFP, arrowheads).

until homokaryosis is confirmed; v) elimination of the unstable pTEL by transfers on nonselective media. This strategy was tested first with pTEL-Fen and Cas9-RNP targeting *Bos1* to generate k.o. mutants via NHEJ. Compared to high transformation rates obtained with pTEL-Fen alone, only few Fen[R] colonies were obtained in initial experiments with 0.5–2 μg pTEL-Fen added together with RNP (6 μg Cas9 complexed with 2 μg sgRNA) to the protoplasts. The apparent suppression of pTEL-Fen transformation rates by Cas9-RNP was largely overcome by increasing the amount of pTEL-Fen in the transformation mixture up to 10 μg (S8 Fig). When Fen[R] transformants were transferred to Ipr containing medium, 25–53% (average 40.0 ± 11.2%) of them proved to be Ipr[R], which demonstrated a high rate of Cas9-RNP mediated editing. After two transfers on nonselective medium, 22 of 26 Ipr[R] transformants had turned to be Fen[S], confirming the expected loss of pTEL-Fen in most cases. Four Ipr[R] transformants remained Fen[R] after further passages, indicating integration of pTEL-Fen into the genome.

In the next cotransformation experiment with pTEL-Fen, *sod1* encoding the major copper/zinc superoxide dismutase was targeted to generate a *sod1-GFP* knock-in fusion (Fig 5C to 5F). Sod1 has been shown to be involved in oxidative stress tolerance and virulence of *B. cinerea* [39]. Several thousand Fen[R] transformants were obtained in single experiments (S8 Fig). Microscopic evaluation revealed GFP fluorescence in 65.3% of the transformants resulting from editing. To verify correct integration of the RT, which eliminated the *sod1* stop codon and attached in-frame the coding sequence of GFP, the complete genome of one fluorescent transformant, and the RT integration borders of three other transformants obtained by PCR were sequenced. In all strains, seamless integration of the RT was confirmed. Ten fluorescent transformants were analyzed by microscopy. In all transformants, fluorescence was observed in the cytoplasm and in strongly fluorescent punctate structures tentatively identified as peroxisomes (Fig 5D). For SOD1 of rat, an orthologue of the fungal Sod1, a localization similar to *B. cinerea* has been found in the cytoplasm and in peroxisomes, due to its binding to peroxisomal protein CCS [40]. The structure of the Sod1-GFP fusion protein was confirmed by immunoblotting using GFP antibodies, which revealed in two transformants bands of approximately 40 kDa, in agreement with the predicted size of Sod1-GFP of 43 kDa, compared to 25 kDa of GFP expressed by a *B. cinerea* control strain (Fig 5E). Functionality of the Sod1-GFP fusion protein was demonstrated by staining for SOD activity after native gel electrophoresis of protein extracts. Whereas in *B. cinerea* WT a major band was visible at about 65 kDa, this band was missing in the Sod1-GFP strain, and instead a larger band was observed. The apparent sizes of these bands were difficult to correlate with the molecular masses of Sod1 (17 kDa) and Sod1-GFP, which could indicate that they formed oligomers (Fig 5F).

Furthermore, cotransformation with pTEL was shown to be useful for marker-free deletion of *nep1* and *nep2*, two genes encoding necrosis and ethylene-inducing proteins [41], by targeting their coding sequences with RNPs and replacing them with a markerless stuffer sequence (Fig 5B). With single targeting, >1,000 Fen[R] transformants were obtained, and 17-23% of these contained *nep1* or *nep2* deletions, respectively, as confirmed by PCR. Co-targeting of *nep1* and *nep2* resulted in 230 transformants. Of these, 12.9% contained a *nep1* deletion and 10% a *nep2* deletion (S9 Fig), but no double transformants were detected. Sequencing of RT integration sites of two *nep1* and two *nep2* transformants, and the genome sequence of one *nep2* transformant (nep2-10) confirmed the expected deletions.

## Telomere vector-mediated marker-free editing also works efficiently in *Magnaporthe oryzae*

The rice blast fungus *M. oryzae* is of great economic importance and considered as the prime model plant pathogenic fungus [2]. It is a hemibiotroph and well-known for its ability to

develop enormous turgor pressure in appressoria facilitating penetration of host cells [42, 43]. Recently, protoplast transformation with CRISPR/Cas using RNP has been successfully applied for this fungus. In order to introduce mutations without a selectable marker, a coediting strategy by cotransformation of a selectable marker gene was also developed, however, rates of non-selected coediting events ranged only from 0.5 to 1.2% [24]. Aiming to improve this rate of marker-free editing in *M. oryzae*, we first confirmed the efficacy of CRISPR/Cas with Cas9-RNP. *M. oryzae* strain Guy11 or Guy11ku80 (a NHEJ deficient mutant) protoplasts were transformed with Cas9-SV40$^{x4}$ complexed with sgRNA *MoALB1* and a Hyg$^R$ RT with about 50 bp homology flanks. *MoALB1* encodes a polyketide synthase required for melanin biosynthesis and *alb1* mutants are easily identified due to whitish mycelium. Depending on the amount of RT DNA used, 67 to 91% of Hyg$^R$ transformants had white mycelium, indicating successful inactivation of *MoALB1*. Next, we tested the suitability of the pTEL-based marker-free approach for *M. oryzae*. After establishing selection for Fen$^R$, using 30 ppm Fen, pTEL-Fen was transformed successfully into *M. oryzae*, yielding up to 1,000 transformants per μg DNA (S2 Table). Subsequently, pTEL-Fen was cotransformed with Cas9-sgRNA RNP targeting *MoALB1*. Among Fen$^R$ transformants, 36–49% displayed white colonies in Guy11, indicating a high editing rate (Fig 6A and 6B; S2 Table). Sequencing of *MoALB1* of three white Fen$^R$ colonies of Guy11 revealed the presence of NHEJ-induced single base pair insertions at the cleaving site, leading to frameshifts, whereas in three white Fen$^R$ colonies of Guy11ku80, deletions of 140 and 342 bp were detected (S10 Fig). After two passages on non-selective medium, 12 out of 15 Guy11 albino mutants were Fen$^S$, as predicted from the instability of pTEL-Fen. In order to demonstrate that editing by insertion of a RT into a specific locus is possible as well, pTEL-Fen, Cas9-RNP targeting the gene *MoPIT* (MGG_01557), and a Hyg$^R$ cassette with 50 bp *MoPIT* homology flanks used as RT were cotransformed into *M. oryzae* protoplasts (S11A Fig). In four independent experiments with Guy11 as a recipient, between 23 and 49 out of 72 Fen$^R$ transformants were Hyg$^R$ (average 49.3%), whereas with Guy11ku80 as a recipient, a lower average (19.1%) of Fen$^R$ transformants were Hyg$^R$ (Fig 6C; S11B Fig). Among the Fen$^R$ Hyg$^R$ transformants, approximately 20% were confirmed as *MoPIT* knockouts by PCR analysis, and the majority of these mutants were the result of homologous integration of the RT, as verified by sequencing of 5' and 3' integration border regions (S11B and S11C Fig). From these data, an editing rate of 4.9% was determined for *M. oryzae* Guy11, and a rate of 1% for Guy11ku80 (Fig 6D). The lower frequency of RT-mediated editing of Guy11ku80 mutant compared to WT was unexpected, since KU80 is known only to be required for NHEJ but not for HR.

## Randomized amino acid editing of a fungicide resistance codon and *in vivo* selection

Succinate dehydrogenase inhibitor (SDHI) fungicides have emerged as the fastest increasing class of fungicides for the control of plant diseases in recent years [4]. Target site mutations leading to resistance against SDHI have been described in *B. cinerea* and other fungi. Most of them are located in *sdhB* encoding the B-subunit of the succinate dehydrogenase enzyme (complex II), an essential component of mitochondrial respiration. In *B. cinerea* populations from SDHI-treated fields, *sdhB* mutations leading to H272R, H272Y, H272L, and H272V amino acid exchanges have been found [44]. While all of them confer resistance to boscalid (Bos), only H272L and H272V mutations also confer resistance to another SDHI, fluopyram (Flu) [4, 45]. To analyze the effects of all possible exchanges in codon 272 for SdhB function and resistance against SDHIs, pTEL-mediated marker-free editing was performed to target *sdhB* with a RT mixture encoding all 20 amino acids in codon 272 (Fig 7A; S3 Table). Several

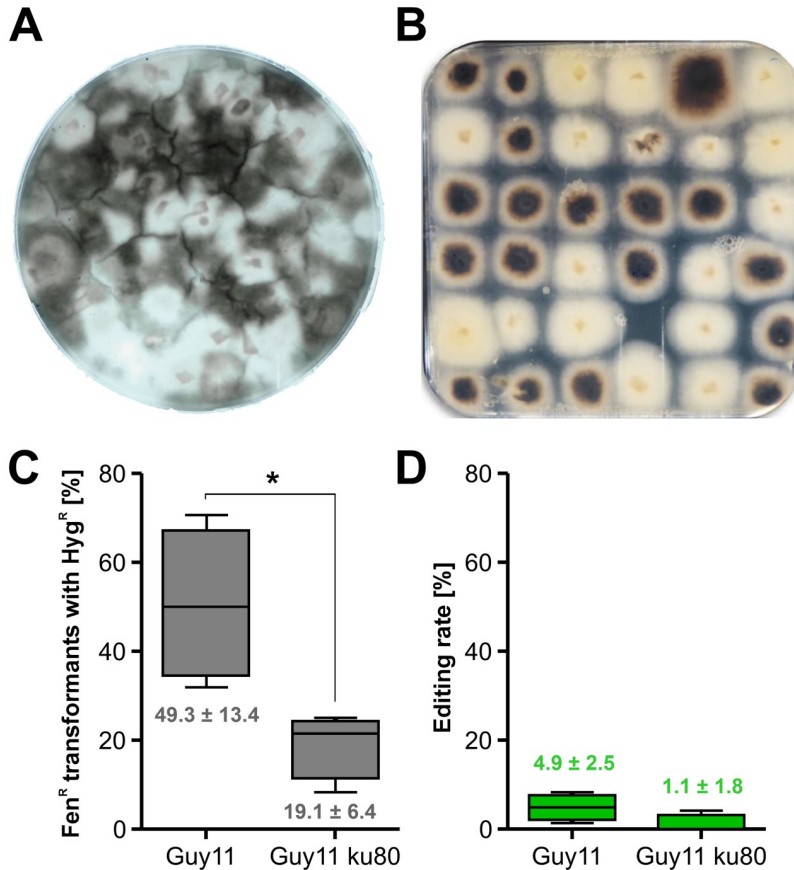

**Fig 6. Efficient pTEL-mediated editing via NHEJ or HR in *Magnaporthe oryzae*.** (A) and (B) Guy11 protoplasts were cotransformed with pTEL-Fen and Cas9-MoALB1-sgRNA RNP. (A) Primary selection plate containing Fen. (B) Isolated transformants (cf. S2 Table). Note white-colored mycelia of edited transformants. (C) and (D) Guy11 and Guy11ku80 protoplasts were cotransformed with pTEL-Fen, Cas9-MoPIT-sgRNA RNP and Hyg$^R$-*MoPIT* RT. (C) Fraction of Fen$^R$ transformants which were also Hyg$^R$. (D) Editing rate, defined as the fraction of Fen$^R$ transformants with correct edits via HR. Values of Guy11 and Guy11ku80 were compared by unpaired t-test. $^*p \leq 0.05$ (n = 4).

thousand colonies were obtained per transformation. Two different sgRNAs were used to lower the risk of choosing one with low editing efficiency. Estimations based on PCR analysis of single transformants revealed editing frequencies between 12.5 and 41%. Average editing frequencies were higher with sgRNA sdhB272-2 than with sgRNA sdhB272-1 (S4 Table). The distribution of codons in position 272 was determined from pooled conidia of $\geq$6,000 Fen$^R$ transformants per assay by bulked DNA isolation, followed by deep sequencing. Aliquots of pooled transformant conidia were cultivated for three days in liquid medium containing discriminatory concentrations of Bos, Flu, or the new SDHI fungicide pydiflumetofen (Pyd) [46], to select transformants with SDHI resistance. DNA of these cultures was isolated and sequenced as above. Among the edited transformants grown on SH+Fen plates, all 20 codons were represented at similar frequencies (Fig 7B). Because in this procedure edited cells may still carry a WT copy of *sdhB* (heterokaryons), we cannot conclude yet that they all maintained full enzyme function. However, our results demonstrate that all H272 amino acids variants yield functional SdhB proteins that are not intrinsically toxic since they all similarly maintained growth and sporulation. Cultivation of the Fen$^R$ transformants in SDHI-containing media followed by quantification of the alleles enabled an unbiased assessment of amino acid

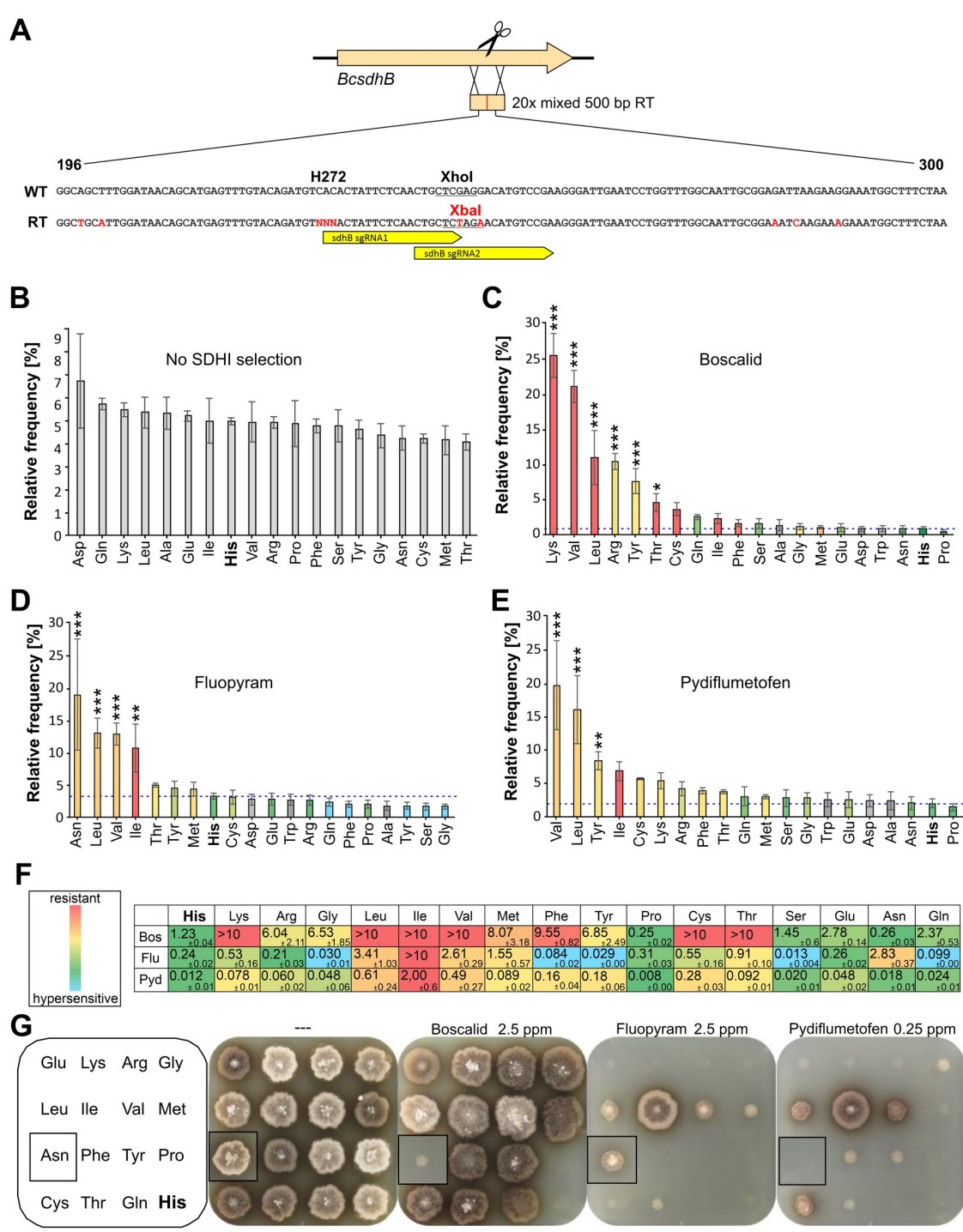

**Fig 7. Effects of *B. cinerea sdhB* codon 272 amino acid randomization by multiple editing.** (A) Schematic strategy, showing sequences of WT and repair template (RT) surrounding codon 272. Changed bases and the new restriction site in the RT are marked in red. NNN: Each of 20 codons in the RT mixture. (B-E) Frequency distribution of encoded amino acids in codon 272 of edited *B. cinerea* transformants, determined by deep sequencing of conidia obtained from primary Fen^R transformants without SDHI fungicide selection (B), or from cultures of transformed conidia incubated in YBA medium containing 0.25 mg l^-1 Bos (C), 0.5 mg l^-1 Flu (D), or 0.15 mg l^-1 Pyd. (E) Fungicide sensitivity levels conferred by each amino acid, as determined for the corresponding mutant are indicated by colors (in (C-F)), except for the bars of amino acids for which no mutants were obtained which are shaded in gray. The *p* values by one-way ANOVA followed by Dunnett's multiple comparisons (control: His) post hoc test are indicated. *$p \leq 0.05$, **$p \leq 0.01$, ***$p \leq 0.001$ (n = 3 or 4). (F) SDHI sensitivity (EC$_{50}$ values in mg l^-1) of individual mutants containing

different amino acids in *sdhB* codon 272. (G) Growth of individual codon 272 edited mutants on YBA agar containing different SDHI as indicated, after 5 days. The inserts show pictures of an Asn mutant which have been pasted into the pictures of the square-sized plates containing the other mutants.

exchanges conferring resistance. Most conspicuous resistance mutations were H272K/V/L/R/Y for Bos, H272N/L/V/I for Flu, and H272V/L for Pyd (Fig 7C–7E).

To analyze the effects of individual amino acid exchanges in *sdhB* codon 272, individual transformants were isolated and the genomic region corresponding to the 500 bp RT was sequenced. By this means, edited strains with 17 different exchanges in codon 272 were identified. Purification of the strains by single spore isolation was continued, until homokaryosis was verified by sequencing. To confirm correct integration of the RT in *sdhB*, the genome sequencing data of four edited strains (sdhB272-Leu, -Met, -Tyr, -Val) were inspected (S12 Fig). Strains sdhB272-Leu and sdhB272-Tyr were confirmed as pure edited strains, whereas sdhB-Met was a heterokaryon with similar abundance of edited and wild type sequences. Strain sdhB272-Val showed an unexpected increase in read coverage over a 500 bp region corresponding to the mixed RT (S12A Fig). Furthermore, while editing of codon 272 was complete, mixed reads were observed in this strain for three more distant divergent nucleotide positions that differed between WT and RT sequences (S12B Fig). This observation can be explained by two events that had occurred in sdhB272-Val: a) Homologous integration of a RT with codon272-Val, but excluding the region containing the three divergent nucleotides, and b) ectopic integration of the complete codon272-Val RT in an unknown region. Purified, homokaryotic edited strains were tested for sensitivity to the three SDHIs (Fig 7F). Overall, the $EC_{50}$ values correlated well with their prevalence in the SDHI selected populations described above (Fig 7C and 7D). Remarkably, 12 amino acids conferred high levels of resistance to boscalid ($EC_{50}$ values $>2$ mg l$^{-1}$) (Fig 7C). In contrast, only four amino acids conferred similarly high resistance levels to Flu, but five amino acids caused up to 30-fold hypersensitivity compared to WT (Fig 7D). Pyd was about ten times more active than Bos and Flu against *B. cinerea* WT, and four amino acids conferred $EC_{50}$ values $>0.2$ mg l$^{-1}$ (Fig 7F). Only the three aliphatic amino acids leucine, valine and isoleucine provided high or intermediate resistance to all three SDHIs. Remarkably, highest resistance levels were observed with isoleucine, which has never been found in resistant field isolates. Growth on selective agar media illustrated the high proportion of Bos$^{R}$ mutants, and the lower number of mutants resistant to Flu and Pyd (Fig 7G). Growth on rich medium and on a nutrient-limited medium with different carbon sources did not reveal significant differences between the 17 edited strains (S13 Fig), indicating no major effects of the amino acids on the fitness of the mutants during vegetative growth.

## Search for off-target mutations in Cas9-RNP-generated *B. cinerea* mutants

A major concern about the use CRISPR/Cas for genome editing is the possibility of off-target mutations at sites which share sequence similarities with the target site. Even if sgRNA sequences are chosen that avoid highly similar secondary regions in the genome, off-site integrations have been observed in sites with several mismatches [47]. Nevertheless, in studies performed with *M. oryzae* and *A. fumigatus* genome editing with Cas9-RNP did not seem to induce off-target effects [24, 48], and did not increase the mutation frequency of transformants compared to standard mutagenesis procedures [48]. To evaluate the occurrence of Cas9-RNP-mediated off-site integrations and the overall frequency of transformation-associated off-target mutations in *B. cinerea*, genome sequencing was performed with the following strains: B05.10 (WT reference), a mutant generated by standard targeted mutagenesis (*nep2*-NatR), a

*bot2boa6* double mutant, three transformants with pTEL-FenR, and six mutants generated by pTEL-mediated marker-free editing. Surprisingly, only very few mutations were observed. Most of these mutations were found to be mixed with WT sequences in the edited strains, apparently due to different degrees of heterokaryotic states of these multinuclear *B. cinerea* strains, despite single spore isolation and confirmation of homokaryosis (S5 Table). The numbers of mutations ranged from one (Sod1-GFP, sdhB272-Leu) to four (*botboa4*-6, pTEL-Fen-8). Altogether, the mutations were single nucleotide exchanges (SNPs, 23 cases) or small insertions or deletions (indels, 7 cases). The SNPs led in six cases to amino acid exchanges in predicted proteins. Nine SNPs and all indels were in intergenic regions or in a telomeric region. These data showed that transformation with pTEL vectors, standard mutagenesis, and CRISPR/Cas with RNPs all resulted in very low frequencies of unwanted mutations in *B. cinerea*. Inspection of the mutated regions did not reveal any sequence similarities to the target sequences of the sgRNA that were used for generation of the edited strains, suggesting that none of these mutations had occurred due to off-target activities of Cas9-RNPs.

## Discussion

Within a short time, CRISPR/Cas genome editing has been used for the genetic manipulation of a wide range of organisms, offering new perspectives in functional genomics. In fungi, advanced CRISPR/Cas systems have been mainly established for *Aspergillus* spp. [34, 49, 50] and *Ustilago maydis* [22, 51]. They take advantage of autonomously replicating circular plasmids, namely the *Aspergillus*-derived AMA1 plasmid and pMS7 in *U. maydis*, for the delivery of Cas9 and sgRNA. AMA1 has also been used in other fungi, including the plant pathogens *Alternaria alternata* [52] and *Fusarium fujikuroi* [53]. However, this plasmid displays only low transformation rates in *B. cinerea* (S. Fillinger, personal communication). Alternatively, a non-integrating vector with human telomeres [28, 54] has been developed in this study as a tool for marker-free editing in *B. cinerea* and *M. oryzae*.

This is the first report of powerful use of CRISPR/Cas for genome editing in *B. cinerea*. A crucial step was the generation of a fully functional, nuclear targeted Cas9. SV40 NLS has been used frequently [22], but efficient nuclear targeting of Cas9-GFP-NLS has been confirmed only in some fungi [25, 55] or optimal activity experimentally verified [53]. In *B. cinerea*, efficient Cas9 nuclear targeting was achieved with C-terminal tandem arrays of either SV40 (4x) and *stuA* (2x) NLS sequences. In most fungi, CRISPR/Cas activity was detected in pilot studies by targeting genes for the biosynthesis of melanin [24, 49]. Following the strategy reported for *Fusarium graminearum* [56], *Bos1* was established as an effective selectable marker for NHEJ- and HR-mediated mutagenesis (Figs 1 and 3). Introduction of Cas9-sgRNA RNPs with or without a donor template yielded hundreds to thousands of edited *B. cinerea* transformants. So far, similar approaches have been rather rarely used for CRISPR/Cas genome editing in fungi [23–25]. An advantage of the use of RNPs over endogenous Cas9 expression is the reduced probability of potential off-target mutagenic activities of Cas9, because of its limited stability in the transformed cells [57]. This was confirmed by genome sequencing of several Cas9-RNP derived transformants, which revealed that Cas9-RNPs do not increase the mutation rate and do not seem to have a significant off-targeting activity in *B. cinerea*. Indeed, we found only one to four mutations in 11 sequenced *B. cinerea* transformants. This is in contrast to traditionally and with Cas9-RNPs generated mutants of *Aspergillus fumigatus*, which contained between 300 and 400 mutations [23], and to Cas9-RNP mutants of *M. oryzae* which contained 74–77 SNP or indel mutations [24]. We don't know the reason for these striking differences in transformation-induced mutations in these fungi. Another advantage of using RNPs is the relatively simple *in vitro* sgRNAs synthesis protocol, which does not require any

cloning steps. Furthermore, RNPs can be tested for their functionality *in vitro* before using them for transformation.

A total of 153 NHEJ repair events in the *Bos1* gene were analyzed, which is the largest number reported for filamentous fungi. Most changes were 1–2 bp indels, and for three sgRNAs inducing a T↓N cleavage by Cas9, a (+T) insertion was the dominating mutation. Although all these mutations were biased by the selection for loss of *Bos1* function (Ipr$^R$), these data are in line with systematic studies of CRISPR/Cas-NHEJ mutations in human cells and yeast [58, 32] which often resulted in +1 bp insertions at the Cas9-RNP cleavage site. This rather reproducible NHEJ repair in *B. cinerea* could be exploited to introduce predictable frameshift mutations even without RT. Furthermore, we demonstrated that RTs with 60 bp homology flanks worked efficiently in *B. cinerea*, yielding >90% targeted integrations (Fig 3). Such short flanks can be attached to a resistance cassette of choice using long PCR primers, avoiding time-consuming cloning or amplification steps which were previously required to generate the long homology flanks for conventional targeted integration.

In *B. cinerea*, cotransformation occurred with rates of up to >60%, both for different combinations of CRISPR/Cas-induced integrations (HR/HR or HR/NHEJ) and for telomere vector uptake and CRISPR/Cas events (HR or NHEJ). Cotransformation rates were found to increase with higher DNA concentrations, consistent with early reports for fungi [59]. Two novel strategies have been established for marker-free editing. Resistance marker shuttling at a non-essential locus in combination with non-selected CRISPR/Cas events allows repeated genomic edits. High frequencies (65.3%) of marker replacement were observed in the transformants, and this approach is also applicable for other organisms. A prerequisite for successful cotransformation and editing in multinuclear fungi such as *B. cinerea* is the generation of homokaryons. First of all, this requires integration of different DNA fragments into the same nuclei. In most transformants analyzed in our study, this was found to be the case, similar to previous reports for *Neurospora crassa* [60]. Secondly, transformants need to be purified by a series of two to five rounds of single spore isolations to achieve homokaryosis, which remains a tedious process even when using the high-efficiency CRISPR/Cas protocols described here. In this respect, marker shuttling provides the advantage to allow screening for the loss of the exchanged marker during purification, as a way to monitor if homokaryosis has been achieved.

The most powerful approach for marker-free editing is cotransformation of a pTEL vector and CRISPR constructs. Its effectiveness depends on i) high transformation efficiency of pTEL which provides the selection, ii) high rates of cotransformation/editing of pTEL and CRISPR components, iii) highly efficient HR and/or NHEJ, and iv) elimination of pTEL after identification of the desired editing event(s), yielding edited strains without any other genomic alterations. With this approach, we reproducibly obtained hundreds to thousands of transformants, and up to 65% of them were marker-free edits. Similar results were obtained for NHEJ- and HR-induced edits, as shown for NHEJ-mediated mutagenesis of *Bos1*, RT-mediated knock-in attachment of a GFP tag to *sod1*, and deletion of *nep1* or *nep2*. Importantly, we could show that pTEL-mediated editing also works well with another filamentous fungus, indicating that it can be applied in many fungi. pTEL-Fen transformed *M. oryzae* protoplasts with equal efficiency as *B. cinerea*, and editing frequencies were 36–49% for NHEJ, and 5% for RT-mediated HR. These values clearly exceed editing rates previously reported with integrative selected markers [24]. The lower rate of editing in *M. oryzae* with HR is probably due to the intrinsically lower efficiency of HR compared to *B. cinerea*. Possibly, this could be partially compensated by using RT with longer homology flanks. We therefore expect that cotransformation with pTEL vectors will significantly facilitate the establishment of RNP-based editing in many fungi, and maybe also in non-fungal microbes such as oomycetes. Given the high

efficiency of CRISPR/Cas-mediated targeting, the frequency of double editing was lower than expected. In the *bot2 boa6* double editing experiments, only three out of 20 double resistant transformants were confirmed to contain the desired mutations (S6 Fig), and in the pTEL-mediated double editing experiment with *nep1* and *nep2*, no double edits were identified. These data indicate that there might be an increased rate of ectopic integrations of RT due to as yet uncharacterized interference effects if multiple RNPs and RTs are transformed. In these experiments, transformation rates (*bot2 boa6*) and editing efficiencies (*nep1 nep2*) were lower than in most other experiments. High transformation rates and editing efficiencies, and careful analysis of the transformants are therefore mandatory for reliably obtaining multiple editings.

The power of pTEL-mediated marker-free editing was exploited by performing an unbiased directed mutagenesis of codon 272 of *sdhB* encoding the succinate dehydrogenase B subunit, the gene in which most mutations conferring resistance against SDHI fungicides have been observed in *B. cinerea* field isolates [4]. Among sporulating transformants, edited strains with all amino acid substitutions were generated with similar frequencies, compared to only four changes detected in field isolates. Drastic differences were observed for the effects of each amino acid on the sensitivity or resistance to the three SDHI tested, which underlines the importance of the conserved histidine 272 for SDHI binding [61]. The majority of substitutions caused high levels of resistance to Bos, whereas fewer substitutions conferred similar resistance levels to Flu and Pyd. Our results are consistent with the observation that Bos resistant *B. cinerea* field isolates with H272R and H272Y substitutions were sensitive or even hypersensitive to Flu and still controllable by this SDHI [44]. Previously, phenotypic characterization of field isolates and of isogenic H272R, H272Y and H272L strains generated by conventional mutagenesis with simultaneous introduction of a resistance cassette at the *sdhB* target locus indicated that these substitutions caused fitness defects, such as aberrant growth and differentiation and reduced competitiveness [62–64]. Although our analysis of the edited strains did not include enzyme activity assays, their equal distribution upon primary selection and normal growth behavior on different media does not support this conclusion. Indeed, this might reflect the great advantage of precise marker-free genome editing in avoiding any modification of neighboring genes or their regulatory sequences by co-introduction of a nearby resistance cassette. Another benefit of our approach is that it allows the analysis of several independent mutants that have been obtained without selection of the target locus. This might obviate the need for tedious complementation experiments to verify the connection between mutations and phenotypes. We further show how selection post-mutagenesis can enable the rapid scanning of mutations conferring resistance to various SDHI fungicides. Since a vast set of target mutations and fungicides can be tested, this new capability is of major relevance for accelerated fungicide design. Interestingly, several substitutions conferring high resistance levels, such as H272I, H272C and H272T have not yet been detected in field populations and suggest that a bias prevented their appearance and propagation in nature. Fungicide resistance mutations are often caused by single nucleotide exchanges, for example the major mutations against most systemic fungicides in *B. cinerea* [3, 65], including the exchanges H272R/Y/L in SdhB [66]. An obvious explanation for their unequal occurrence is their differential effects on fungal fitness, therefore mutations resulting in minimal fitness costs are most likely to occur [67]. Our data, showing similar SDHI resistance and fitness levels caused by hitherto unknown substitutions seem to indicate that fungicide resistance development in field populations is also limited by the number and probability of mutations required to change one codon to another [68].

The high yield of telomere vector-mediated editing in combination with RNP-CRISPR/Cas opens the door to advanced genome editing applications with *B. cinerea* and other fungi, such as large-scale mutagenesis and gene tagging projects. Approaches similar to mutagenesis of

*sdhB* codon 272 are now possible for *in vivo* selection and structure-function analysis of proteins, such as those involved in fungicide resistance, host invasion or any other functions of interest.

## Materials and methods

### Fungi

*Botrytis cinerea* B05.10 was used as WT strain in this study. For demonstration of CRISPR/Cas-assisted marker replacement, a *B. cinerea* B05.10 derivative, containing a Nat$^R$ cassette (PtrpC-*nat*-TniaD) integrated in *xyn11A* [38] was used. Cultivation of *B. cinerea* and infection tests were performed as described [6]. Guy11 was used as *Magnaporthe oryzae* WT strain. A NHEJ-deficient *M. oryzae* mutant, Guy11ku80, was kindly provided by A. Foster.

### DNA constructs for transformations

All oligonucleotides used are listed in S6 Table. Sequences of plasmids marked with * are provided in S1 Text: Plasmid sequences. Derivates of the telomere vector pFAC1 [28] were constructed as following: pFAC1 was digested with BglII/NheI, and the vector fragment ligated with a synthetic linker made by annealing of oligonucleotides pFAC1-del1/pFAC1-del2, resulting in pFB2N*, carrying a hygromycin resistance cassette. For telomere vector-mediated editing, a truncated version of pFB2N carrying a fenhexamid resistance marker was generated, called pTEL-Fen*. A codon optimized version of the *Streptococcus pyogenes cas9* gene for expression in *B. cinerea*, under the control of *oliC* promoter from *A. nidulans*, was synthesized by Genewiz (South Plainfield, NJ, USA). To generate a stable Cas9 expressing *B. cinerea* strain, a nourseothricin resistance cassette consisting of *A. nidulans trpC* promoter (PtrpC), *nat* gene and *B. cinerea gluc* terminator (Tgluc) [15] was integrated next to the *cas9* gene, and homology flanks for targeted integration of the construct into *niaD* encoding nitrate reductase were added. For efficient nuclear localization of Cas9, a synthetic sequence encoding four copies of the SV40 T antigen NLS (SV40$^{x4}$) was C-terminally attached to the *cas9* coding sequence, resulting in pUC-BcCas-SV40x4_nat_niaD*. To test different NLS arrangements for their efficiency to target Cas9 into nuclei of *B. cinerea*, Cas9 was fused to GFP codon-optimized for *B. cinerea* (from pNAH-OGG [15]) and the following NLS sequences C-terminally attached: Single copy SV40, SV40$^{x4}$, Stu$^{x2}$ (a tandem duplicated NLS of Bcin04g00280 encoding a homologue of the *A. nidulans* nuclear protein StuA [52]), and SV40$^{x2}$ (each one N- and C-terminal SV40). For transient expression of Cas9-GFP, pFB2N was first truncated by digestion with BlpI/SphI, followed by ligation with the annealed oligonucleotides FB108/ FB109, resulting in pFB2N_BlpI_MreI. This plasmid was digested with BlpI/MreI and ligated with fragments containing Cas9-GFP-NLS, resulting in pTEL-BcCas9GFP-NLS-SV40x4* and pTEL-BcCas9GFP-NLS-Stux2*.

To generate a RT with 1 kb *Bos1* homology flanks and a fenhexamid resistance cassette, pBS-KS(-) was digested with EcoRV and combined by Gibson assembly with two adjacent 1 kb *Bos1* homology flanks, using primers TL29 pBS_ol_bos 3.REV/ TL30 pBS_ol_bos 3.FOR and TL31 pBS_ol_bos 1.FOR/ TL32 pBS_ol_bos 1.REV, and a fenhexamid resistance cassette amplified from pNDF-OCT [30] with primers TL33 Fen_ol_bos 2.FOR/TL34 Fen_ol_bos 2.REV. From the resulting plasmid (pBS_Bos1_KO_Fen), *Bos1* RT with short homology flanks were amplified with the following primers: TL37_Fen_fw/ TL38_Fen_rev (0 bp); TL65_Bos1_Fen30_fw/ TL66_Bos1_Fen30_rev (30 bp), TL67_Bos1_Fen40_fw/ TL68_Bos1_Fen40_rev (40 bp); TL69_Bos1_Fen60_fw/ TL70_Bos1_Fen60_rev (60 bp). A RT with 60 bp *Bos1* homology flanks at 1 kb distance from the cleavage site was amplified from pTEL-Fen using primers TL113 60bp Bos1 PD FW/ TL114 60bp Bos1 PD RV. For generation of *boa6* k.o. mutants, a

newly designed cyprodinil resistance cassette was used, which was based on the use of a cyprodinil resistant version of the *Bcpo5* gene containing a L412F mutation [65].

## Expression of Cas9 protein with *B. cinerea* optimized NLS

SV40[x4] and Stu[x2] NLS were fused to the 3'-terminus of *Streptococcus pyogenes* Cas9 (*E. coli* codon optimized) and cloned into pET24a. The resulting plasmids, pET24a_Cas9-SV40x4-NLS-His* and pET24a_Cas9-Stux2-NLS-His* were used to express these Cas9 derivatives in *E. coli* BL21(DE3) at 20˚C in autoinduction medium. Cells were harvested and ca. 10 g of cell paste resuspended in 50 ml extraction buffer (20 mM HEPES, 25 mM imidazole, 500 mM NaCl, 0.5 mM TCEP, pH 8) by stirring for 40 min. Cells were lysed using a Cell Disruptor (Constant Systems Limited, Daventry, UK) at 20,000 psi, and the lysate clarified by centrifugation at 20,000 rpm in a fixed-angle rotor for 30 min, 4˚C. The lysate was applied to a 5 ml HisTrap FF column equilibrated in extraction buffer. Bound protein was eluted with 3.5 column volumes of elution buffer (20 mM HEPES, 500 mM imidazole, 500 mM NaCl, pH8, 0.5 mM TCEP). The eluate was loaded onto a GE 26/60 S200 SEC column equilibrated in 20 mM HEPES, pH8, 0.5 mM TCEP. Fractions containing the target protein were pooled and 20% (v/v) glycerol was added. The solution was concentrated using a 10 kDa Vivaspin column. Aliquots were frozen in liquid nitrogen and stored at -80˚C until use. Functionality of *in vitro* assembled Cas9-sgRNA complexes was tested by *in vitro* cleavage of target DNA as described (S3C Fig) [69]. For comparison, a commercially available version of Cas9, containing SV40 NLS on both termini (New England Biolabs Inc., Beverly, MA, USA) was used.

## Synthesis of sgRNA and RNP formation

Selection of appropriate sgRNAs was carried out with the help of the sgRNA design tool of the Broad Institute (https://portals.broadinstitute.org/gpp/public/analysis-tools/sgrna-design). Oligonucleotides for synthesis of sgRNAs are listed in S5 Table. DNA template preparation was performed by annealing 10 μmol each of constant sgRNA oligonucleotide (TL147_gRNA rev) and protospacer specific oligonucleotide in 10 μl in a thermocycler (95˚C for 5 min, from 95˚C to 85˚C at 2˚C sec$^{-1}$, from 85˚C to 25˚C at 0.1˚C$^{-1}$), followed by fill-in with T4 DNA polymerase (New England Biolabs, Beverly, MA, USA), by adding to the annealing mix 2.5 μl 10 mM dNTPs, 2μl 10x NEB buffer 2.1 (50 mM NaCl, 10 mM Tris-HCl, 10 mM MgCl$_2$, 100 μg ml$^{-1}$ BSA, pH 7.9), 5 μl water and 0.5 μl enzyme, incubation for 20 min at 12˚C and column purification. Subsequently, sgRNA synthesis was performed using the HiScribe T7 High Yield RNA Synthesis Kit (NEB), and purified using the RNA Clean & Concentrator-25 kit (Zymo Research, Orange, CA, USA). Cas9-NLS, containing N- and C-terminal SV40 NLS, was purchased from NEB. For RNP formation, 6 μg Cas9 was incubated in cleavage buffer (20mM HEPES, pH 7.5, 100 mM KCl, 5% glycerol, 1 mM dithiothreitol, 0.5 mM EDTA, pH 8.0, 2 mM MgCl$_2$) with 2 μg sgRNA for 30 min at 37˚C. The amounts of gRNA were determined by spectrophotometry, and their quality estimated by denaturing 10% polyacrylamide gel electrophoresis, using TBE buffer (89 mM each of Tris base and boric acid, 2 mM sodium EDTA) and 8 M urea, and TBE as running buffer.

## Transformation of *B. cinerea*

Transformation was performed based on a published protocol [6] as following: $10^8$ conidia harvested from sporulating malt extract (ME: 10 g/l malt extract, 4 g/l glucose, 4 g/l yeast extract, pH 5.5) agar plates were added to 100 ml ME medium and shaken at 180 rpm for ca. 18 h (20–22˚C) in a 250 ml flask. The germlings were transferred into 50 ml conical tubes and

centrifuged (8 min, 1,000 g) in a swing-out rotor. The combined pellets (fresh weight should be >3 g) were resuspended and washed two times with 40 ml KCl buffer (0.6 M KCl, 100 mM sodium phosphate pH 5.8; centrifugation for 5 min, 1,000 g), and the germlings resuspended in 20 ml KCl buffer containing 1% Glucanex (Sigma Aldrich, St Louis, MO, USA; L1412) and 0.1% Yatalase (Takara, T017), and incubated on a 3D rotary shaker at 60 rpm for 60–90 min at 28˚C until ca. $10^8$ protoplasts had been formed. Protoplasts were filtered through a sterile nylon mesh (30 μm pore size) into a 50 ml conical tube containing 10 ml ice-cold TMS buffer (1 M sorbitol, 10 mM MOPS, pH 6.3). After addition of another 30–40 ml ice-cold TMS buffer, the suspension was centrifuged (5 min, 1500 g, 4˚C), and the protoplast pellet resuspended in 1–2 ml TMSC buffer (TMS + 50 mM $CaCl_2$, 0˚C, dependent on the desired protoplast concentration. To $5x10^6$ to $2x10^7$ protoplasts in 100 μl TMSC, the Cas9/sgRNA ribonucleoprotein (RNP) complex (6 μg Cas9, 2 μg sgRNA; pre-complexed for 30 min at 37˚C) and up to 10 μg donor template DNA were added in 60 μl Tris-$CaCl_2$ buffer (10 mM Tris-HCl, 1 mM EDTA, 40 mM $CaCl_2$, pH 6.3). After 5 min incubation on ice, 160 μl of PEG solution (0.6 g ml$^{-1}$ PEG 3350, 1 M sorbitol, 10 mM MOPS, pH 6.3; pre-heated to 60˚C, mixed, and allowed to cool down to 30–40˚C) was added, mixed gently, and incubated for 20 min at room temperature. 680 μl of TMSC buffer was added, the sample was centrifuged (5 min, 1,500 g in a swing-out rotor), the supernatant removed, and protoplasts suspended in 200 μl TMSC. Protoplasts were transferred into 50 ml liquid (42˚C) SH agar (0.6 M sucrose, 5 mM Tris-HCl pH 6.5, 1 mM $(NH_4)H_2PO_4$, 9 g l$^{-1}$ bacto agar, Difco) and poured into two Petri dishes. For transformation with pTEL-Fen, up to 10 μg plasmid DNA was used. For selection of transformants, 30 mg l$^{-1}$ nourseothricin (Nat), 1 mg l$^{-1}$ fenhexamid (Fen), 4 mg l$^{-1}$ iprodione (Ipr), or mg l$^{-1}$ fludioxonil (Fld) were added. Positive colonies were transferred onto ME agar plates or onto plates containing the same concentrations of selective agents. Transformants were subcultured on selective media and purified by three to five rounds of single spore isolation. Total DNA for PCR analysis was isolated as described previously [70]. DNA for genome sequencing of *B. cinerea* was isolated by grinding ca. $5x10^7$ ungerminated conidia in liquid nitrogen using a mortar and pestle, followed by the yeast protocol of the Blood & Cell Culture DNA Kit Mini (Qiagen, Hilden, Germany). For genome sequencing, DNA input was normalized to 15ng/μl. Sequencing libraries were prepared with 30ng of normalized DNA using the NEBNext Ultra II FS DNA Library Prep Kit for Illumina (New England BioLabs, Ipswich, Massachusetts, USA). Custom 8 bp barcode was added to each library during the preparation process. Sample were then pooled together, and pool was cleaned with magnetic beads included in the library preparation kit. Pool was the run on a lane of the HiSeqX instrument (Illumina, San Diego, CA, USA) in a 150-cycle paired end run. Total sequence yield was 150 Gb, which resulted in a range of 8–21 Gb for each sample.

## Transformation of *M. oryzae*

Three-day old cultures of *M. oryzae* Guy11 or the Guy11ku80 deletion mutant, grown in 150 ml liquid complete media at 25˚C and 100 rpm, were used for generation of protoplasts. The mycelia were filtered and digested with Glucanex as described above [71]. Protoplast purification was done according to the protocol for *B. cinerea*. After washing with TMS buffer, protoplasts were suspended in TMSC buffer and adjusted to $1.5x10^8$ protoplasts per ml. For transformation, 120 μl aliquots of a protoplast suspension were mixed with the RT DNA and pre-incubated RNPs (Cas9-SV40$^{x4}$) dissolved in 60 μl Tris-$CaCl_2$ buffer. Then 180 μl 60% PEG 3350 were added, and the protoplast suspension was poured into CM agar containing 1.2 M sucrose for osmotic stabilization. After 24 h an upper layer containing 500 mg l$^{-1}$ hygromycin (Hyg) or 30 mg l$^{-1}$ Fen was poured over the agar containing the protoplasts. After 7–10 days,

mutants were transferred to selection plates for further selection. RT (containing *gpd3* promotor, *hph* and *tubB* terminator) with 50 bp of homology flanks was amplified using primers MH-Alb F&R for targeting *MoAlb1*, and MH-Pit F&R for targeting *MoPIT*. For sgRNA synthesis, primers sgRNA_Alb1 and sgRNA_Pit were used. Transformants were verified using primers SeqPit F/R, SeqAlb F/R and MoPit FL F/R.

### Generation and *in vivo* selection of *sdhB* codon 272 edited strains

To be used as mixed RT for randomized editing, twenty 500 bp *sdhB* fragments differing in codon 272 (listed in S4 Table) were synthesized by Twist Bioscience (San Francisco, U.S.A.) and pool-amplified with primers TL148_ SDHB_RT_F/ TL149_ SDHB_RT_R. Illumina deep sequencing was performed to verify equal representation of each fragment (±7.5%). For PCR-based identification of edited transformants, silent mutations were introduced into the 500 bp fragments which converted an XhoI to an XbaI site (codons 278/279), and allowed differentiation between WT and edited sequences (Fig 7a). To isolate the DNA of *sdhB* codon 272-edited transformants for sequencing, sporulation was induced on the primary transformation plates. For this, three days after transformation, the SH+Fen agar containing embedded transformants was overlaid with 0.1 volumes of 5x concentrated ME medium. After another 5–7 days, transformant conidia were harvested from densely sporulating plates. To improve the recovery of transformants, the agar discs were inverted, placed onto fresh ME (1 mg l$^{-1}$ Fen) agar plates, and incubated again for 5–7 days until sporulation. Conidia harvested from one transformation were combined and used for DNA isolation. For sequence analysis of bulked transformants selected for resistance to SDHI, 4x10$^5$ conidia of Fen$^R$ transformants were inoculated in standard Petri dishes with 18 ml YBA medium (1% yeast extract, 20 g l$^{-1}$ sodium acetate [72]) containing boscalid (0.25 mg l$^{-1}$; BASF, Ludwigshafen, Germany), fluopyram (0.3 mg l$^{-1}$; Bayer Crop Science, Monheim, Germany), or pydiflumetofen (0.015 mg l$^{-1}$; Syngenta Crop Protection, Stein, Switzerland) in concentrations inhibitory for *B. cinerea* WT strain B05.10. After 72 h incubation at 20˚C, conidia and germlings were harvested and used for DNA isolation [70] and sequencing (see below).

To isolate *sdhB* edited strains with defined codon 272 replacements, individual Fen$^R$ transformants were purified by several transfers on ME+Fen (1 mg l$^{-1}$), YBA+Bos (1 mg l$^{-1}$), or YBA+Flu (2.5 mg l$^{-1}$) agar media. Total DNA of these isolates was amplified using primers TL151_SDHB_OS_F/ TL152_SDHB_OS_R, and the 741 bp products digested with either XbaI or XhoI to test whether they were edited or WT. Edited isolates were sequenced using primer TL148_ SDHB_RT_F or TL149_ SDHB_RT_R.

### Sequencing and analysis

For deep amplicon sequencing of edited transformants, bulked *B. cinerea* DNA was first amplified in 20 μl total volume, with 2 μl DNA, 10 pM of primers sdhb_F1/ sdhb_R1, 1x MyTaq buffer, and 1 Unit MyTaq (Bioline; Meridian Bioscience Inc., London, UK) by incubation for 2 min at 96˚C, followed by 20 cycles of 15s 96˚C, 30s 60˚C, 90s 70˚C. Nested PCR was performed in 20 μl total volume, using 2 μl of the first round PCR, under the same conditions as above, but with 15 cycles only. PCR products were purified with AmpureXP beads (Thermo Fisher Scientific, Bremen, Germany). About 100 ng of each purified PCR product was used to construct Illumina libraries using the Ovation Rapid DR Multiplex System 1–96 (NuGen Technologies, San Carlos, CA, USA). Illumina libraries were pooled and size selected by preparative gel electrophoresis. Sequencing (3 million reads per sample) was performed by LGC Genomics (Berlin, Germany) on an Illumina NextSeq 550 instrument with v2 chemistry in 2x150 bp read mode. Libraries were demultiplexed using Illumina's bcl2fastq 2.17.1.14

software. Sequencing adapter sequences were removed from the 3' end of reads with cutadapt (https://cutadapt.readthedocs.io/en/stable/) discarding reads shorter than 20 bp. All read pairs were filtered for valid primer combinations and reverse-complemented so that R1 corresponds to the forward primer and R2 to the reverse primer. Actual primer sequences were removed for downstream processing. Reads were quality-filtered by LGC proprietary software, removing all reads with an average Phred score below 30, and all reads containing more than 1 undetermined base (N). Subsequently, all read pairs were overlap-combined using BBMerge 34.48 from the BBMap package (https://jgi.doe.gov/data-and-tools/bbtools/bb-tools-user-guide/bbmerge-guide/). Mutated positions were identified by a custom shell script, filtering for sequences containing the motifs immediately before and after these mutated triplet (TTTGTACAGATGT and ACTATTCTCAACTG, respectively). The sequence content between these motifs were extracted and counts for the detected sequences summarized for each sequencing library.

For full genome sequencing of *B. cinerea* strains, gDNA was normalized to 15 ng $\mu l^{-1}$. Sequencing libraries were prepared with 30 ng of normalized DNA using the NEBNext Ultra II FS DNA Library Prep Kit for Illumina (New England BioLabs, Ipswich, Massachusetts, USA). Custom 8 bp barcode was added to each library during the preparation process. Sample were then pooled together. The pool was cleaned with magnetic beads included in the library preparation kit, and run on a lane of the HiSeqX instrument (Illumina, San Diego, CA, USA) in a 150-cycle paired end run. Total sequence yield was 150 Gb. This resulted in a range from 8–21 Gb for each sample. The B05.10 reference genome and annotation gtf file were created by combining the nuclear genome (ASM83294v1), mitochondrial genome (GenBank: KC832409.1) and the plasmid pTEL-Fen. Reads were quality checked using FastQC version 0.11.4 [73] and multiqc [74] and trimmed using Trimommatic version 0.39 [75] to remove adapters and low quality bases. Trimmed reads were aligned to the reference genome using bwa mem version 0.7.17 [76]. SAM and BAM files were manipulated using Samtools version 1.7 [77]. Duplicated reads were marked using Picard Tools version 2.20.2 [78]. Variants were called using Samtools mpileup and bcftools version 1.9 and filtered for a read depth of 10 and quality score of >20. Variants were then annotated using SnpEff version 4.1g [79]. Chimeric and discordant reads were identified using Samtools. Sequencing of the twelve *B. cinerea* strains resulted in genome coverages between 189x (sdh272-Met) and 498x (sdhB272-Leu), with an average of 228 ± 95x. The fraction of ≥Q30 bases was between 92.24 and 93.62.

## Fungicide susceptibility test

Isolates with defined edits in codon 272 were tested for radial growth on YSS (2 g $KH_2PO_4$; 1.5 g $K_2HPO_4$; 1 g $(NH_4)_2SO_4$ x $7H_2O$; 2 g yeast extract) agar with 50 mM each of either glucose, malate, acetate or succinate [63], and for their sensitivities to SDHIs. Susceptibility to Bos (BASF), Flu (Bayer Crop Science), and Pyd (Syngenta) was assessed in the WT and in edited strains on the basis of inhibition of germination. Assays with a range of fungicide concentrations (0, 0.001, 0.003, 0.01, 0.03, 0.1, 0.3, 1, 3, 10 mg $l^{-1}$) were carried out in YBA medium (1% yeast extract, 20 g $l^{-1}$ sodium acetate) at 20°C. After incubation for 30 h in Greiner Bio-one polystyrene microtiter plates, the fraction of conidia containing germ tubes with lengths exceeding half of the conidial diameters was determined for each strain/fungicide pair, and an $EC_{50}$ value (effective fungicide concentration required to inhibit germination by 50%) was calculated with the Graphpad Prism 5.01 software, using a normalized response with variable slope fitted to log fungicide concentrations.

## Microscopy

Confocal images were acquired using either a Leica SP5 (DM6000 CS), TCS acousto-optical beam splitter confocal laser scanning microscope, equipped with a Leica HCX PL APO CS 63 × 1.20. water-immersion objective or a Zeiss LSM880, AxioObserver SP7 confocal laser-scanning microscope, equipped with a Zeiss C-Apochromat 40x/1.2 W AutoCorr M27 water-immersion objective. Fluorescence signals of GFP (Leica: excitation/emission 488 nm/500-550 nm, Zeiss: excitation/emission 488 nm/500-571 nm), were processed using Leica software LAS AF 3.1, Zeiss software ZEN 2.3 or Fiji software.

## Protein analysis

For in-gel detection of superoxide dismutase (SOD) activity, *B. cinerea* conidia were germinated in ME medium overnight, washed with extraction buffer (100 mM potassium phosphate buffer (pH 7.8) 0.1 mM (EDTA) 1% (w/v) polyvinyl-pyrrolidone (PVP) 0.5% (v/v) Triton X 100), and the mycelium ground using mortar and pestle in liquid nitrogen. The frozen powder was suspended in extraction buffer and centrifuged for 20 min at 4˚C at 15,000 g. Fifteen μg of clear supernatant were separated on a native polyacrylamide gel. SOD activity was detected by inhibition of reduction of nitro blue tetrazolium by $O_2^-$. SOD-active areas became visible as clear zones on a blue-violet background. [80]. For detection of Cas9 or Sod1-GFP fusion proteins, *B. cinerea* protein extracts prepared as described above were separated in SDS polyacrylamide gels and subjected to an immunoblot on nitrocellulose, using monoclonal antibodies against Cas9 (Clontech, Palo Alto, CA, USA) or GFP (Sigma), followed by chemiluminescent detection.

## Statistics and reproducibility

Statistical analyses were carried out with the GraphPad Prism software. The detailed analysis method is depicted in the individual figure legends. All experiments were carried out at least three times. For growth and infection assays, three technical replicates per sample were performed. Box limits of box plots represent 25th percentile and 75th percentile, horizontal line represents median. Whiskers display minimum to maximum values. Bar charts represent mean values with standard deviations.

## Supporting information

**S1 Fig. Detection of Cas9 expression in *B. cinerea*.** Total *B. cinerea* protein extracts (15 μg per lane) were loaded, separated by polyacrylamide gel electrophoresis, and Cas9 detected with a monoclonal Cas9 antibody. M: Marker; 1: B05.10 (WT); 2: B05.10-Cas9-SV40$^{x4}$ (stably integrated gene); 3: B05.10 (pTEL-Cas9-Stu$^{x2}$).
(TIF)

**S2 Fig. Sensitivity to osmotic and salt stress and virulence of *B. cinerea* WT and CRISPR/Cas-induced *Bos1* mutants.** (A) Pictures of three Ipr$^R$ *Bos1* mutants (M98, M99, M100, all having the same '+T' insertion) and WT growth for 48 h on ME medium without (—) and with 0.5 M NaCl or sorbitol. (B) Effects of salt and osmotic stress treatments on radial growth, compared to growth on pure ME medium (n = 3). The *p* values by one-way ANOVA followed by Dunnett's multiple comparisons (control: WT) post hoc test are indicated. $^{**}p \leq 0.01$; $^{***}p \leq 0.001$ (n = 4). (C) Infection on tomato leaf by WT and mutant M99 (72 h).
(TIF)

**S3 Fig. *In vitro* and *in vivo* performance of RNPs with different sgRNAs targeting *Bos1*.**
(A) Positions and expected cleavage sites (red dotted lines) of the sgRNAs (yellow) in *Bos1*.
PAM sequences for each of the sgRNAs are indicated in red. (B) Summary of on-target scores,
*in vitro* cleavage activities, and transformation efficiencies with different sgRNAs. *On-target
efficiency scores calculated with the Broad Institute GPP sgRNA Designer. ***In vitro* cleavage
efficiency was estimated from gel pictures. *** Number of Ipr$^R$ *B. cinerea* transformants per
assay. (C) Examples of *in vitro* cleavage reactions with sgRNAs bos1-T1, -T2 and -T3.
(TIF)

**S4 Fig. Sequence analysis of Cas9-RNP-induced cleavage-repair sites in *Bos1*, obtained
from Ipr$^R$ transformants.** (A) Ethidium bromide-stained agarose gels (negative pictures) with
PCR fragments generated with primers TL_87 Bos1_check 200 Fw/ TL_87 Bos1_check 200
Rv, showing variations of fragment sizes due to different types of NHEJ-induced mutations in
transformants obtained with Cas9/bos1-T1 RNP. (B) Origins and sequences of different types
of NHEJ insertions obtained with Cas9/bos1-T1, Cas9/bos1-T2 and Cas9/bos1-T3 RNPs. Type
A (not shown): 164 bp *B. cinerea* mitochondrial DNA, two joined fragments of 84 and 79 bp.
Type B: *B. cinerea Bos1*-DNA. Type C: 15 bp fragment of the sgRNA scaffold encoding part of
the T7 RNA polymerase promoter. Type D: sgRNA scaffold DNA containing part or all of the
protospacer sequences of bos1-T1/-2/-3. Type E: sgRNA scaffold DNA lacking protospacer
sequences. Positions of the sgRNAs relative to the *Bos1* sequence (in green) are shown. compl:
Insertion in inverse orientation, complementary sequence is shown. The bases flanking the
Bos1-T3 sgRNA cleavage site are indicated with a blue background.
(TIF)

**S5 Fig. Transformation of Cas9/*Bos1*-T2B-gRNA RNP and two Fen$^R$ RT with long *Bos1*
homology flanks.** (A) Experimental scheme. *Bos1* inactivation leading to Ipr$^R$ occurs either by
targeted integration of the Fen$^R$ RT via HR, or via NHEJ. As RT, either a circular plasmid or a
PCR fragment amplified from this plasmid, both containing 1 kb *Bos1* homology flanks, were
used. (B) Transformation results: Primary selection was either for Ipr$^R$ (white bars) or for Fen$^R$
(grey bars) (n = 3). When the PCR fragment or the plasmid were transformed without RNP as
control, no Fen$^R$ colonies, except for one colony in one experiment, were obtained. Below the
diagram, the fraction of transformants with resistance to both fungicides is shown. (n): Num-
ber of transformants tested. Statistical analyses were performed by analysis of variance
(ANOVA, followed by Dunnett's multiple comparisons. No significant differences between
transformation results with PCR fragments and circular plasmids, or between the numbers of
Ipr$^R$ and Fen$^R$ colonies obtained with the same batches of transformed protoplasts were
observed.
(TIF)

**S6 Fig. Cas9-RNP-mediated single and double k.o. mutagenesis of *bot2* and *boa6*.** (A)
Transformation results. (B) PCR-based verification of *bot2 boa6* double (ΔΔ) k.o. mutants.
*boa6* RT right flank (RF) integration screen using primers TniaD_ol_Cyp_Fw/ TL129 (537
bp); *boa6* WT screen using primers TL157/ TL158 (263 bp), *bot2* RT RF integration screen
using primers TL130/ TL132 (444 bp), *bot2* WT screen using primers TL133/ TL159 (180 bp).
+: *B. cinerea* WT DNA (positive control); -: no template DNA (negative control). (C) Growth
of WT and mutants after 72 h on agar plates with rich (ME) and minimal (GB: Gamborg B5
with 25 mM glucose) medium (one-way ANOVA; n = 3). (D) Lesion formation after 72 h on
tomato leaves (one way ANOVA; n = 3). In (C) and (D), no significant differences in radial
growth and infection between WT and mutants were observed.
(TIF)

**S7 Fig. Confirmation of loss of the kan^R gene in *B. cinerea* transformed with pTEL-Fen.**
(A) Map of pTEL-Fen, indicating the PCR fragments that were amplified. (B) Negative picture of ethidium bromide stained agarose gel showing the result of the PCR reactions performed with total DNA from the *B. cinerea* strains used for sequencing. Expected sizes were for kan^R: 625 bp; PtrpC: 504 bp. (C) Alignment of Illumina bam reads mapping to the linearized sequence of pTEL-Fen in the DNA of *B. cinerea* strain pTEL-Fen8.
(TIF)

**S8 Fig. Transformation results for pTEL-mediated *Bos1* k.o. via NHEJ, depending on the amounts of pTEL-Fen added to protoplasts.** Individual data points are shown. The p value by one-way ANOVA followed by Dunnett's multiple comparisons (control: 0.5 μg pTEL-Fen) post hoc test is indicated. $^*p \leq 0.05$.
(TIF)

**S9 Fig. Use of pTEL-Fen for marker-free deletion of *nep1* and *nep2*.** (A) Transformation result. (B) PCR-based verification of *nep1* deletion mutants, using primers TL143/ TL144; size of WT fragment 1,353 bp, size of *nep1* k.o. fragment 733 bp. Lanes 1–5: Fen^R transformants. Transformant #5 represents a nearly pure *nep1* mutant. (C) PCR-based verification of *nep2* deletion mutants, using primers TL145/ TL146; size of WT fragment 1,220 bp, size of *nep2* k.o. fragment 641 bp. Lane 6: *B. cinerea* WT; lanes 7–10: Fen^R transformants. Transformant #10 represents a purified *nep2* mutant. M: DNA marker.
(TIF)

**S10 Fig. Sequencing of Cas9-RNP generated, pTEL-Fen mediated *ΔMoalb1* mutants.** (A) Alignment of a *M. oryzae ALB1* genomic sequence section (WT strain Guy11) and three edited mutants showing a white phenotype. A DNA fragment around the *MoALB1*-sgRNA binding site (shown as reverse complement: rc ALB1-sgRNA) was amplified with primers SeqAlb1 F/ SeqAlb1 R, and the PCR product sequenced. (B) Alignment of a *MoALB1* genomic sequence (strain Guy11ku80) section with the sequences of three edited mutants showing a white phenotype. Amplification and sequencing were done as described above. Alignments were done with Jalview 2.10.4.
(TIF)

**S11 Fig. pTEL-mediated gene replacement of *MoPIT* via HR in *M. oryzae*.** (A) Schematic illustration of *MoPIT* locus with sgRNA and binding sites for primers used in this study. The upper cartoon delineates the genomic region of *MoPIT*. The middle cartoon shows the repair template (RT) containing Hyg^R and 50 bp flanks on each site for homologous recombination. The cartoon at the bottom displays the mutated *MoPIT* region after gene replacement. (B) Results for co-transformations of strains Guy11 and Guy11ku80 with pTEL-Fen, Cas9-sgRNA RNP and Hyg^R-*MoPIT* repair template. Four independent experiments were performed with isolate Guy11ku80 (a-d) and Guy11 (e-f). After transformation, protoplasts were firstly selected for Fen^R, and 72 of the mutants were additionally tested for Hyg^R. This revealed 8–25% for Guy11ku80 and 32–68% for Guy11 mutants being resistant to both antibiotics. PCR analyses were applied to test mutants, co-resistant to Fen^R and Hyg^R, for the presence of the repair template at the *MoPIT* locus (see part C). From them 18% for Guy11ku80, however only in one out of four experiments, and 4–25% for Guy11, showed integration of the Hyg^R-*MoPIT* repair template. To discriminate whether gene replacement was due to HR or NHEJ sequencing of PCR products, amplified with primers SeqPit5'_F/ SeqPit5'_R, and primers SeqPit3'F/ SeqPit3'R, was performed. Thus, all three Guy11ku80 mutants showed the predicted gene replacement by HR while the frequency for Guy11 varied between 50–80%. (C) PCR-based verification of pTEL-mediated gene replacement was done for all mutants co-selected

for Fen[R] and Hyg[R] (exemplary pictures are shown for mutants from experiment a, mutants M1-M23). PCR amplification was done using primers MoPit FL F and MoPit FL R. Thus, amplification of the *MoPIT* WT gene yielded a PCR-product size of 683 bp while the band size corresponding to successful gene replacement was 1835 bp. The presence of two PCR products, at 683 bp and 1835 bp (M18, M21), indicated an unspecific integration of the complete Hyg[R] construct into the genome without gene replacement. Mutants showing HR-mediated *MoPIT* replacement (M5, M6, M10, M16, M19) are marked with asterisks (\*).
(TIF)

**S12 Fig. Bam-file reads showing sequence coverage and mutations in the *sdhB* region.** (A) Read coverage in the region surrounding *sdhB*. Note duplicated coverage in the region corresponding to the 500 bp *sdhB* repair template in sdhB272-Val. (B) Mutations in the region surrounding codon 272. Note mixed edited/ WT reads indicating heterokaryosis in strain sdhB272-Met, and mixed edited/ WT reads restricted to three positions on the right presumably due to homologous and in addition ectopic integration of the repair template in strain sdhB272-Val.
(TIF)

**S13 Fig. Mycelium growth after 72 h of *B. cinerea sdhB* codon 272 exchange mutants on agar media containing ME or YSS with different carbon sources (50 mM each), relative to the WT strain (n = 3).** Statistical analyses were performed by analysis of variance (ANOVA) followed by Dunnett's multiple comparisons (control: His). No significant differences between the growth rates of the WT strain (His) and any of the mutants were observed by one-way ANOVA followed by Dunnett's multiple comparisons (control: His) post hoc test.
(TIF)

**S1 Table. Instability of pTEL vectors in *B. cinerea* under non-selective culture conditions.** Conidia of *B. cinerea* strains containing vectors pFB2N, pTEL-BcCas9GFP-NLS-Stux2 and pFB2N (stable Hyg[R] strain Bcgfp1 (25) and stable Fen[R] strain niaD-fferg27 as controls) were transferred from solid media containing ME+Hyg and ME+Fen, respectively, to the edge of a non-selective ME plate and allowed to grow and sporulate for 7 d. From this plate, conidia were harvested from a ca. 1 cm$^2$ area above the site of inoculation (site A) and from a similar area at 4 cm distance from the inoculation site. The conidia were transferred at low density to either ME+Hyg or ME+Fen plates. After 20 h incubation, the germination rate from 100–500 spores was determined. n.a.: not analysed.
(XLSX)

**S2 Table. pTEL-mediated editing in *MoALB1* via NHEJ in *M. oryzae*.** CRISPR/Cas components used and results of (co-) transformations with strain Guy11 and Guy11ku80. Transformations with the same letter were done with the same batch of protoplasts.
(XLSX)

**S3 Table. *sdhB* codon 272 mixed amino acid repair template sequences (20 x 500 bp).**
(XLSX)

**S4 Table. Results of cotransformation of *B. cinerea* with pTEL-Fen, sdhB272-sgRNA-RNP and 500bp sdhB-272 (x20) repair template.** [1] 2x10$^7$ protoplasts were transformed with 10μg pTEL-Fen, 6μg Cas9-2μg sdhB272-RNP, and 10μg *sdhB*-272(x20) repair template. [2]To determine the fraction of edited transformants, DNA of individual transformants was prepared, amplified with primers TL151_SdhB_OS_F/ TL152_SdhB_OS_R covering the edited region, and digested with either XhoI (cut in WT *sdhB* DNA) or XbaI (cut in edited *sdhB* DNA). n.a.:

not analysed.
(XLSX)

**S5 Table. Off-target mutations in transformed and Cas9-RNP edited *B. cinerea* strains.**
The table shows positions which were covered by $\geq$10 clean sequence reads in the B05.10 reference (WT) and all other strains, and which contained $\geq$10% altered reads (Mutation) in one of the transformed strains. The numbers indicate the fractions of altered reads. [*1]: T(AGAGG GAG)$_6$A(GA)$_3$GGGA(GGGAGAGA)$_2$GGGAGA; [*2]: T(AGAGGGAG)$_6$AGAGG(GA)$_5$(GG GA)$_2$(GAGAGGGA)$_2$GA [*3]: A(AAAAAAAG)$_5$; [*4]: A(AAAAAAAG)$_3$.
(XLSX)

**S6 Table. Oligonucleotides used.**
(XLSX)

**S1 Text. Plasmid sequences.**
(DOCX)

**S1 Data. Raw data.**
(XLSX)

## Acknowledgments

We are grateful to Sabine Fillinger (INRA, Paris, France) for providing telomere plasmid pFB2N, and to Pinkuan Zhu (Shanghai Normal University, China) for help with pTEL constructions. We thank Patrick Pattar for excellent technical support, and Nora Fischbach for help with characterization of edited strains. We thank Lucio Garcia (Molecular analytics, Syngenta US) and Stephanie Widdison (General bioinformatics, Syngenta UK) for their support with full genome sequencing and analysis thereof. Andrew Foster is kindly acknowledged for providing Guy11ku80.

## Author Contributions

**Conceptualization:** Gabriel Scalliet, Matthias Hahn.

**Data curation:** Gabriel Scalliet, Matthias Hahn.

**Formal analysis:** Thomas Leisen, Fabian Bietz, Janina Werner, Alex Wegner, David Scheuring, Felix Willmund, Andreas Mosbach.

**Funding acquisition:** Ulrich Schaffrath, Gabriel Scalliet, Matthias Hahn.

**Investigation:** Thomas Leisen, Fabian Bietz, Janina Werner, Alex Wegner, David Scheuring, Matthias Hahn.

**Methodology:** Thomas Leisen, Fabian Bietz, Matthias Hahn.

**Supervision:** Ulrich Schaffrath, Matthias Hahn.

**Writing – original draft:** Matthias Hahn.

**Writing – review & editing:** Ulrich Schaffrath, Felix Willmund, Gabriel Scalliet, Matthias Hahn.

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
