## [Decision Letter · Decision Letter 0]

23 Feb 2020

Dear Dr. Hahn,

Thank you very much for submitting your manuscript "CRISPR/Cas with ribonucleoprotein complexes and transiently selected telomere vectors allows highly efficient marker-free and multiple genome editing in Botrytis cinerea" for consideration at PLOS Pathogens. As with all papers reviewed by the journal, your manuscript was reviewed by members of the editorial board and by several independent reviewers. In light of the reviews (below this email), we would like to invite the resubmission of a significantly-revised version that takes into account the reviewers' comments. We would like to note that particularly the concerns of referee 2 deserve careful attention and require additional evidence to support the claims that are made. We cannot make any decision about publication until we have seen the revised manuscript and your response to the reviewers' comments. Your revised manuscript will also be sent to reviewers for further evaluation.

Sincerely,

Melvin Bolton

Guest Editor

PLOS Pathogens

Bart Thomma

Section Editor

PLOS Pathogens

Kasturi Haldar

Editor-in-Chief

PLOS Pathogens

orcid.org/0000-0001-5065-158X

Michael Malim

Editor-in-Chief

PLOS Pathogens

orcid.org/0000-0002-7699-2064

Reviewer's Responses to Questions

**Part I - Summary**

Reviewer #1: Leisen et al. present a novel adaptation of CRISPR/Cas9 technology to the necrotrophic plant pathogen Botrytis cinerea. The major advancement presented in this manuscript is the utilization of a stable telomere vector to enable transient selection of CRISPR/Cas9 mediated edits without integration of the selectable marker. This protocol was expanded to the important plant pathogen Magnaporthe oryzae and was shown to function with high efficiency. This methodology was also used to test various types of genome edits (knock-in of a fluorescent protein, gene deletions), but also used to assess the effects of all possible amino acid substitutions in codon 272 of the sdhB gene. Phenotypic effects of each mutation were evaluated and novel insights into the SDHI resistance mechanisms were obtained. Overall, the experiments presented in this manuscript were well designed and controlled, and the data supports the authors’ conclusions. The paper was well-written and generally clear. The methodology developed in this manuscript is a significant achievement for functional gene analyses across many fungal taxa. Additionally, although not the main focus of the paper, the sdhB functional data presented not only highlights the power of this method, but also provides important information regarding fungicide resistance and may have implications in fungicide chemistry development and/or testing.

Reviewer #2: Review PPATHOGENS-D-20-00059

The manuscript presents a potentially interesting new transient selection system that might improve the efficiency of the CrispR-based co-editing strategy previously reported in the rice blast fungus. The authors claim that their system uses ‘autonomous unstable telomere vectors’ to achieve this increase in efficiency. However there is no experimental evidence to support this claim except for the loss of resistance due to culturing the fungal strains without selection.

In a paper that describes a novel experimental approach, it is important that all necessary experimental details are given – for example how long does this curing process take and how exactly is it done (does it require single spore isolation, for instance)? The authors additionally claim that their approach allows ‘introduction of ‘clean’ markerless changes into the genome without unwanted modifications’ and in the discussion they claim the generation of ‘edited strains without any other genomic alterations’ – again there is no experimental evidence to support such a claim and this could not be made without whole genome sequence data to support the assertion of no ‘unwanted modifications’. It is also very important that the authors show that their approach is not more mutagenic than non-CrispR methods.

The experiments made with Botrytis seem to have in the main, well-performed, except that in most of the cases reported the genotype is inferred from the phenotype only. There was an attempt made to prove that the strains assumed to be inactivated for BOS1 really contain indels in the target gene. However there is no genotype analysis made for the bot2- and boa6- transformants, as far as this reviewer could see – it is really necessary to sequence the amplicons in these cases. This is very important.

There are more serious issues with the Magnaporthe transformations which appear to be just reporting one transformation experiment which was not reproduced. The authors must know that an experiment is not reproduced unless repeated exactly, and that performing an experiment with three different concentrations of a constituent does not constitute repetition of the experiment. There is again very limited attempt to genotype the mutants generated - just three strains were analysed (the PCR products were sequenced) according to the manuscript and this information is not actually shown at all. I feel that it is necessary to confirm the assumed genotypes in many more than just three strains. The transformation plate shown in Fig. 6 seems to show confluent growth and I wonder how the authors were able to isolate individual clones from such a dense plating in the absence of an obvious phenotype such as altered pigmentation.

In summary I believe that this work shows a lack of rigour in many places and makes very broad claims based on very little evidence and I do not think it therefore merits publication in this journal. The method described is in actual fact, very largely an adaptation of existing approaches that have now been reported in many species, including notably Magnaporthe oryzae, where a very similar approach was published in 2018.

I would recommend that the authors address the fate of the ‘teeomere-vectors’ used using pulsed field gel electrophoresis and Southern blotting and/or whole genome sequencing before and after curing in both species. This would provide the evidence they really need to support the claims being made. It is also important to explore what happens to the plasmids and whether they are really maintained extra-chromosomally, as assumed by the authors. Additionally, given the fact that a long period of growth with and then without selection is required, I would again recommend that the authors perform whole genome sequencing of some of the strains generated to test whether their approach is more mutagenic than non-CrispR based gene targeting methods.

The authors should also describe the past Magnaporthe CrispR approaches, more thoroughly in their introduction – it should be made clear that efficient CrispR-based gene targeting approaches have already been reported in the species and that a co-editing approach, as described here, has already been described before. Likewise in fairness to the scientists that developed the telomere targeting vector previously, the literature describing it should be introduced in the introduction. The authors seem to be rather over-claiming their approach, which is at best an incremental advance to other reported methods, using telomere vectors originally generated by others.

The manuscript might be better approached not as a ‘methods paper’, unless more detailed evidence in support of the claims made is provided, but instead by focusing on the application in understanding fungicide resistance development, which is arguably the most novel finding that is reported.

Reviewer #3: The submitted manuscript describes the development and testing of RNP-Cas9 based genome editing in the model fungal plant pathogens Botrytis cinerea and Magnaporthe oryzae. The research describes the use of a transient selection vector, which contains teleomeres for temporary selection without marker integration. Results using this method showed efficient genome editing for a fungicide resistance target gene, allowing the creation of 20 amino acid coding mutants for a particular residue known to contribute to fungicide resistance. The manuscript provides details and describes an approach that will further advance the use of CRISPR-based genome editing in fungi and will likely be of interest to those in the field. It is a nice study.

There are however sections of the manuscript that are not well described or presented that make it difficult to read and understand. These should be addressed before acceptance as a key attribute of this work is to facilitate other labs to replicate the work and system for enhanced genome editing efficiency.

**Part II – Major Issues: Key Experiments Required for Acceptance**

Reviewer #1: (No Response)

Reviewer #2: More specific comments

Fig 2 B – it is necessary to explain exactly what T1 T2 T3 and T4 refer to – this is not clear

Fig 5 F and G – the sizes of the proteins do not correspond between the two panels?

Fig 6 – why are the pigmented transformants green – my understanding was that this fungus is grey (hence its former name)?

Fig S5 – how many times was this experiment repeated (if at all?) – this needs to be explicitly stated for all the experiments made in fact.

Fig S8 – there is really no need to show again CrispR based gene targeting with selection given that 3 publications have already demonstrated that this approach works very well in this species. It would additionally appear that this experiment was only carried out once (?).

Fig S9 A – these experiments were also only done once?

Reviewer #3: Conclusion lines 203-205. Can the authors summarize/quantify the types of edits observed to support the comments of ‘remarkably uniform mutation patterns.’ The description suggests that many different types of edits were recovered.

Fig. 1

In 1E, what are the # of transformants relative to the N=4, 4, and 11. Also, it is not clear what the statistical test is comparing? Multiple comparison to what? In E, Tukeys post-hoc is reported, while in F Dunnets testing used? Why the difference? For Dunnetts, what is compared against?

Fig. 2

The labeling is unclear between A and B. Which bars in B correspond to guides in A. It should be more explicit.

S5 fig. (lines 208-215)

For Fen insertion with 1kb flank, it does not appear as though there was a control. There is no information provided on the rate of Fen insertion from linear or plasmid without Cas9. With such long homologous flanks (1kb) there could be a significant number of insertions from homologous recombination regardless of Cas9 cutting. This data is not interpretable without controls.

Fig. 3 The grey versus black is very difficult to see and the overall layout is difficult. There needs to be clearer labeling. It was not clear from the narrative why there is such a difference between Ipr and Fen resistance numbers as seen in Fig 3D.

Throughout the text and figure 5, the term co-editing is used, however, it is co-transformation that is described. The expression of Fen from the pTel along with Cas9 editing is not co-editing. Only a single edit is made.

There is no control for 5E, 5F, 5G. What does a random GFP insertion look like? What does GFP expressed alone show on the blots. Is the blot in F and G from the image shown in E? What is the expected size of the Sod1-GFP fusion. Also, the analysis for SOD activity is not common knowledge and results should be explicitly explained for 5F.

Line 313 again does not show co-editing. Only one locus is being edited, the co- is DNA uptake. Line 313-314, no data shown or referenced.

Is the M. oryzae data with the pTEL-Fen vector provided? Is line 314-315 for the pTEL-lines?

S8 Fig has an error for the numbers. 18/18 is not 72%. Also, there are again no controls. No Cas9 alone, sgGuides alone. Also, the ku80 strain is to test for NHEJ versus HR, but there is not an hph insert lacking homology. That is needed to show that the ku80 strain fails to produce any transformants. This lack of a control becomes apparent for the results in fig 9.

The lack of explanation and labeling in S9 makes it very difficult to interpret. What is k.o.s? What are the asterisk for gel image? The reader should not be required to explicate these data/figures. It should be clearly labeled and described. Why are bands in M18 and M21 not mentioned?

How do they know that editing (small INDEL) did not take place in the PCR+ mutants? One interpretation of the KU80ko losing all integration of Hyg is that the integrations of the hyg were not HR but NHEJ insertions. There is no control where the hyg has no homology, so this cannot be ruled out. Also, why is this called coediting (line 320). Only one site is being edited, MoPIT.

Line 411-412. Why is highly efficient HR a requirement? Ku80 data shown in S9 suggests NHEJ is important for DNA integration.

Line 419. It is not clear from data show in fig 6 that the transformants are from NHEJ? The hph donor is not described in the text with the data?

M&M. 516-517, It is not clear what this line is referring too? The functionality of the nucleases was tested? This data should be provided then.

**Part III – Minor Issues: Editorial and Data Presentation Modifications**

Reviewer #1: P5L108: change “telomer” to “telomere”

P8l174: change “cinereal” to “cinerea”

Figure 1: change “Tuckey’s” to “Tukey’s” in the caption

S1 Figure: change “cinereal” to “cinerea”

P9L191-193: edit sentence

Figure 2: It appears that sgRNA Bos1-T2 is missing from part A of the figure (Bos1-T1 is labeled twice).

P10L222: Shouldn’t this be 64% (16/25 transformants without microhomology were both resisant to Fen and Ipr)?

P10L223-224: What were the sequence artefacts introduced during integration of Fen via NHEJ (i.e. was there a prominent insertion/deletion type)?

P12L273-279: What were the rates of confirmed Fen/Ipr double resistant transformants for each of the pTEL-Fen concentrations used in this experiment? The number of transformants appeared to increase with increasing pTEL-Fen concentration, but did the rate of co-editing increase? If these data were recorded, it would be nice to include them here.

P14L333/Figure7: An explanation of why two sgRNAs were used in this experiment would be useful for the audience.

P18L416: Can the author’s provide any speculation as to why nep1/nep2 double knockouts were not obtained?

Reviewer #2: Minor errors

Line 161 and application > an application

Line 182 significantly less should be significantly fewer

Line 186. k.o. mutants is not an appropriate or precise description

Lines 196-197 ‘typical for error-prone NHEJ repair of CRISPR/Cas-induced DNA breaks’ – a reference is required here

Line 414 ‘up to >50%’ – up to 50% or always 50% or more?

Line 562 Protoplasts > Protoplast

Line 911 ‘colonies of an Asn mutant which’ – this is unclear – describe the mutation more precisely

Reviewer #3: Abstract, line 102-103: The term clean is vague, and the marker-less approach does not guarantee unwanted modifications.

Line 117: is the word ‘cultures’ a misspelling? Otherwise, the meaning is unclear.

Line 142-143 descibes HR in the presence of donor sequence. I think it is appropriate to add language that the presence of homologous sequence can or may stimulate HDR, but it is not a given outcome.

Line 418. Fungus, not fungi. Only a single other fungus is shown.

M&M 643-644 is not well described and not clear. Also, what is Cas9 Sod1-GFP fusion proteins?

PLOS authors have the option to publish the peer review history of their article (what does this mean?). If published, this will include your full peer review and any attached files.

Reviewer #1: No

Reviewer #2: No

Reviewer #3: No
---

## [Decision Letter · Decision Letter 1]

3 Jul 2020

Dear Dr. Hahn,

Thank you very much for submitting your manuscript "CRISPR/Cas with ribonucleoprotein complexes and transiently selected telomere vectors allows highly efficient marker-free and multiple genome editing in Botrytis cinerea" for consideration at PLOS Pathogens. As with all papers reviewed by the journal, your manuscript was reviewed by members of the editorial board and by several independent reviewers. The reviewers appreciated the attention to an important topic. Based on the reviews, we are likely to accept this manuscript for publication, providing that you modify the manuscript according to the review recommendations.

Sincerely,

Melvin Bolton

Guest Editor

PLOS Pathogens

Bart Thomma

Section Editor

PLOS Pathogens

Kasturi Haldar

Editor-in-Chief

PLOS Pathogens

orcid.org/0000-0001-5065-158X

Michael Malim

Editor-in-Chief

PLOS Pathogens

orcid.org/0000-0002-7699-2064

Reviewer Comments (if any, and for reference):

Reviewer's Responses to Questions

**Part I - Summary**

Reviewer #3: The authors have addressed the major concerns from the previous version. There are more details provided for the methodology, and the results are more clearly presented. The usefulness of the approach is apparent and should aid in functional genomic studies in fungi.

**Part II – Major Issues: Key Experiments Required for Acceptance**

Reviewer #3: There is still a problem with using the term 'coediting', as detailed in Fig. 5A. There is only a single locus edited by Cas9, therefore it cannot be coedited. The authors acknowledged this point in their response, but there seem to be additional places in the manuscript where this term needs to be modified for accuracy.

**Part III – Minor Issues: Editorial and Data Presentation Modifications**

Reviewer #3: (No Response)

PLOS authors have the option to publish the peer review history of their article (what does this mean?). If published, this will include your full peer review and any attached files.

Reviewer #3: No
---

## [Editor Report · Decision Letter 2]

10 Jul 2020

Dear Dr. Hahn,

We are pleased to inform you that your manuscript 'CRISPR/Cas with ribonucleoprotein complexes and transiently selected telomere vectors allows highly efficient marker-free and multiple genome editing in Botrytis cinerea' has been provisionally accepted for publication in PLOS Pathogens.

Best regards,

Melvin Bolton

Guest Editor

PLOS Pathogens

Bart Thomma

Section Editor

PLOS Pathogens

Kasturi Haldar

Editor-in-Chief

PLOS Pathogens

orcid.org/0000-0001-5065-158X

Michael Malim

Editor-in-Chief

PLOS Pathogens

orcid.org/0000-0002-7699-2064
---

## [Editor Report · Acceptance letter]

4 Aug 2020

Dear Dr. Hahn,

We are delighted to inform you that your manuscript, "CRISPR/Cas with ribonucleoprotein complexes and transiently selected telomere vectors allows highly efficient marker-free and multiple genome editing in Botrytis cinerea," has been formally accepted for publication in PLOS Pathogens.

Best regards,

Kasturi Haldar

Editor-in-Chief

PLOS Pathogens

orcid.org/0000-0001-5065-158X

Michael Malim

Editor-in-Chief

PLOS Pathogens

orcid.org/0000-0002-7699-2064